# Narrow-bandwidth sensing of high-frequency fields with continuous dynamical decoupling

Alexander Stark[1,2], Nati Aharon[3], Thomas Unden [ID] [2], Daniel Louzon[2,3], Alexander Huck [ID] [1], Alex Retzker [ID] [3], Ulrik L. Andersen[1] & Fedor Jelezko[2,4]

State-of-the-art methods for sensing weak AC fields are only efficient in the low frequency domain (<10 MHz). The inefficiency of sensing high-frequency signals is due to the lack of ability to use dynamical decoupling. In this paper we show that dynamical decoupling can be incorporated into high-frequency sensing schemes and by this we demonstrate that the high sensitivity achieved for low frequency can be extended to the whole spectrum. While our scheme is general and suitable to a variety of atomic and solid-state systems, we experimentally demonstrate it with the nitrogen-vacancy center in diamond. For a diamond with natural abundance of $^{13}$C, we achieve coherence times up to 1.43 ms resulting in a smallest detectable magnetic field strength of 4 nT at 1.6 GHz. Attributed to the inherent nature of our scheme, we observe an additional increase in coherence time due to the signal itself.

[1] Department of Physics, Technical University of Denmark, Fysikvej, Kongens Lyngby 2800, Denmark. [2] Institute for Quantum Optics, Ulm University, Albert-Einstein-Allee 11, Ulm 89081, Germany. [3] Racah Institute of Physics, The Hebrew University of Jerusalem, Jerusalem 91904, Israel. [4] Center for Integrated Quantum Science and Technology (IQst), Ulm University, Ulm 89081, Germany. Correspondence and requests for materials should be addressed to A.S. (email: astark@fysik.dtu.dk) or to F.J. (email: fedor.jelezko@uni-ulm.de)

mproving the sensitivity of high-frequency sensing schemes is of great significance, especially for classical fields sensing[1–3], detection of electron spins in solids[4, 5], and nuclear magnetic resonance spectroscopy[6]. The common method to detect high-frequency field components is based on relaxation measurements, where the signal induces an observable effect on the lifetime, $T_1$, of the probe system[4, 5, 7]. Nevertheless, the sensitivity of this method is limited by the pure dephasing time $T_2^*$ of the probe system.

Pulsed dynamical decoupling[8–10] can substantially increase the coherence time[11–18]. In order to carry out sensing with a decoupling scheme, the frequency of the decoupling pulses has to be matched with the frequency of the target field[19, 20]. This largely restricts the approach to low frequencies, as the repetitive application of pulses is limited by the maximum available power per pulse[21]. The same power restrictions are present for very rapid and composite pulse sequences aimed to decrease both external and controller noise[22–26].

With continuous dynamical decoupling (CDD)[21, 27–38] robustness to external and controller noise can be attained, especially for multi-level systems[39–42]. However, the significance of CDD for sensing high-frequency fields remained elusive. Indeed, it was unclear whether it is possible to incorporate such a protection into the metrology task of sensing frequencies in the GHz domain. The first step towards this goal was done recently by integrating CDD in the sensing of high-frequency fields with three level systems[42].

In this article, we propose, analyze, and experimentally demonstrate a sensing scheme that is capable of probing high-frequency signals with a coherence time, $T_2$, limited sensitivity. Unlike relaxation measurements comprising a bandwidth $\propto 1/T_2^*$, determined by the pure dephasing time, $T_2^*$, of the sensor (up to the MHz range), our protocol overcomes the imposed limitation by protecting the addressed two-level system (TLS) with an adapted concatenated CDD approach. We use and adjust it such that high-frequency sensing becomes feasible even for not phase-matched signals. As a result, the proposed scheme is generic and works for many atomic or solid-state TLS, in which the energy gap matches the frequency of the signal under interrogation. A remarkable feature of our scheme is the fact that the signal to be probed also works partially as a decoupling drive and thus improves further the sensitivity of the sensor.

We demonstrate the performance of CDD by applying it to a nitrogen-vacancy (NV) center in diamond with natural abundance of $^{13}$C. Here, we utilize two of its ground sub-levels as the TLS. The states of the NV center can be read out and initialized by a 532 nm laser, which reveals spin-dependent fluorescence between the two levels[43–45]. The system can be manipulated by driving it with microwave fields. We show that by using a concatenation of two drives, an improvement in coherence time of the sensor by more than one order of magnitude is achieved. Taking into account the effect of an external signal, $g$, on the sensor during a concatenation of two drive fields, we obtain an improvement in bandwidth for high-frequency sensing by three orders of magnitude in comparison to the relaxometry approach. Moreover, we report on the measurement of a weak high-frequency signal with strength $g$, which relates to a smallest detectable magnetic field amplitude of $\delta B_{\min} \approx 4$ nT.

## Results

**The sensing scheme.** The basic idea of utilizing concatenated continuous driving to create a robust qubit is illustrated in Fig. 1a, b. The concatenation of two phase-matched driving fields results in a robust qubit[36, 42]. In what follows we show that such a robust qubit can be utilized as a sensor for frequencies in the range of the

qubit's energy separation and hence, dynamical decoupling can be integrated into the sensing task.

By the concatenated driving, the qubit is prepared in a state that allows for strong coherent coupling to the high-frequency signal to be probed (corresponding to the last TLS in Fig. 1c). In the total Hamiltonian, $H$, we consider the concatenation of two driving fields of strength (the Rabi frequency) $\Omega_1$ and $\Omega_2$, respectively. The Hamiltonians of the TLS, $H_0$, the protecting driving fields, $H_{\Omega_1}, H_{\Omega_2}$ and the signal, $H_s$, are given by

$$H = H_0 + H_{\Omega_1} + H_{\Omega_2} + H_s$$

$$= \tfrac{\omega_0}{2}\sigma_z + \Omega_1\sigma_x\cos(\omega_0 t) + \Omega_2\sigma_y\cos(\omega_0 t)\cos(\Omega_1 t) + g\sigma_x\cos(\omega_s t + \varphi),$$

$$(1)$$

where $\omega_0$ is the energy gap of the bare states ($\hbar = 1$), $\omega_s$ is the frequency of the signal, and $g$ is the signal strength which we want to determine. We tune the system, i.e., $\omega_0$, $\Omega_1$, and $\Omega_2$, such that $\omega_s = \omega_0 + \Omega_1 + \Omega_2/2$.

It is an important feature that phase matching between the signal and the control is not required, which means that the signal phase $\varphi$ can be unknown and moreover, it may vary between experimental runs. In addition, we make the assumption that $\omega_0 \gg \Omega_1 \gg \Omega_2 \gg g$. Moving to the interaction picture (IP) with respect to $H_0 = \tfrac{\omega_0}{2}\sigma_z$ and making the rotating-wave approximation, we obtain

$$H_I = \tfrac{\Omega_1}{2}\sigma_x + \tfrac{\Omega_2}{2}\sigma_y\cos(\Omega_1 t)$$

$$+ \tfrac{g}{2}\left(\sigma_+ e^{-i((\Omega_1+\Omega_2/2)t+\varphi)} + \sigma_- e^{+i((\Omega_1+\Omega_2/2)t+\varphi)}\right).$$

$$(2)$$

This picture incorporates the effect of $\Omega_1$ onto a TLS and express the new system in eigenstates of $\sigma_x$, the $|\pm\rangle$ (dressed) states, which separates the contributions from $\Omega_2$ and $g$. For a large enough drive $\Omega_1$, the $|\pm\rangle$ eigenstates are decoupled (in first order) from magnetic noise, $\delta B\sigma_z$, because $\langle\pm|\sigma_z|\pm\rangle = 0$. However, power fluctuations $\delta\Omega_1$ of $\Omega_1$ limit the coherence time of the dressed states. The resulting IP is illustrated in the second TLS in Fig. 1c.

We continue by moving to a second IP with respect to $H_{01} = \tfrac{\Omega_1}{2}\sigma_x$, which leads to

$$H_{II} = \frac{\Omega_2}{4}\sigma_y + \frac{g}{4}\left(-i\sigma_+ e^{-i\left(\frac{\Omega_2}{2}t+\varphi\right)} + i\sigma_- e^{+i\left(\frac{\Omega_2}{2}t+\varphi\right)}\right). \quad (3)$$

Once again, we incorporate $\Omega_2$ into the dressed states, so that solely the contribution of the signal $g$ becomes obvious, which is depicted in the last TLS of Fig. 1c. The second drive, $\Omega_2$, which is larger than $\delta\Omega_1$, creates effectively doubly dressed states (the $\sigma_y$ eigenstates). These doubly dressed states are immune to power fluctuations of $\Omega_1$ and hence prolong the coherence time (see Supplementary Note 2 for more details). Moving to the third IP with respect to $H_{02} = \tfrac{\Omega_2}{4}\sigma_y$ results in

$$H_{III} = \frac{g}{8}\left(\sigma_+ e^{-i\varphi} + \sigma_- e^{+i\varphi}\right), \quad (4)$$

where we can clearly see that the signal $g$ induces rotations in the robust qubit subspace (either with $\sigma_+$ or $\sigma_-$). These rotations are obtained for any value of an arbitrary phase $\varphi$ and the bandwidth ($\propto 1/T_2$) is now limited by the coherence time, $T_2$, of the sensor. Hence, if a given TLS exhibits the possibility of manipulating it via drive fields $H_{\Omega_1}$ and $H_{\Omega_2}$, we can achieve a high-frequency sensor in the range of $\omega_0$.

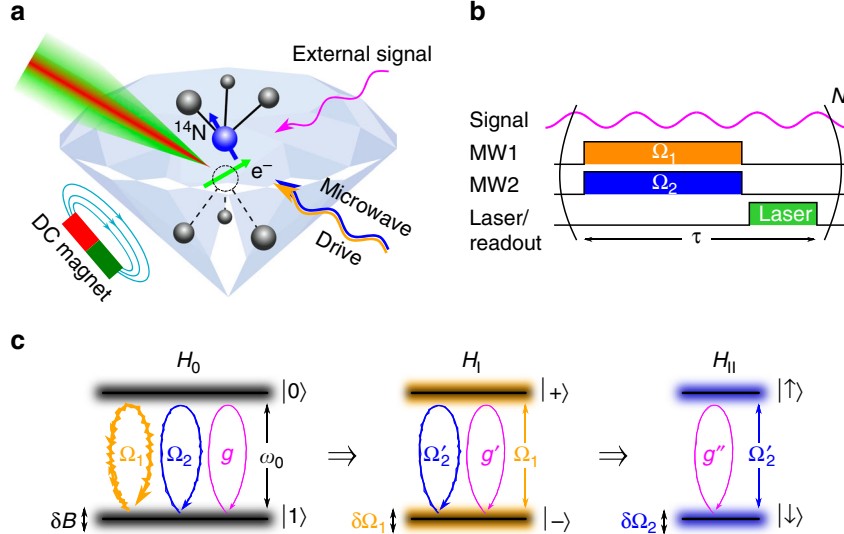

**Fig. 1** Schematic representation of our setup. **a** The NV center probes an external signal while it is being manipulated by the control fields. **b** Schematic representation of the sequence applied in this work. **c** The protected TLS: the bare system, $H_0$, is subjected to strong environmental noise $\delta B$. Applying a strong drive, $\Omega_1$, opens a protected gap, now subjected mainly to drive fluctuations $\delta\Omega_1$. A second drive, $\Omega_2$, is then applied to protect the TLS, $H_I$, from these fluctuations, resulting in a TLS, $H_{II}$, on resonance with the signal, $g'' = g/4$, with noise mainly from the second weak drive $\delta\Omega_2 \ll \delta\Omega_1$

By this, we overcome the low frequency limit that is common to state-of-the-art pulsed dynamical decoupling sensing methods. In addition, we present an analog pulsed version of our scheme, where the pulsing rate is much lower than the frequency of the signal (Supplementary Note 9). However it is not a direct measurement of the signal, but based on a signal demodulation approach. Compared to the pulsed schemes, CDD does not suffer from being susceptible to higher harmonics of the decoupling window appearing naturally from the periodic character of the pulsed sequence[46]. Eventually, less power per unit time is used in the continuous scheme leading to a smaller overall noise contribution from the drive.

**Implementation and analysis of the presented scheme.** After determining the optimal drive parameters, $\Omega_1$ and $\Omega_2$, for the concatenated sensing sequence, and thereby maximize the coherence times, $T_2^{\Omega_1}$ and $T_2^{\Omega_1,\Omega_2}$, respectively, of the sensor (Supplementary Note 6), we apply an external high-frequency signal (according to $H_s$ in Eq. (2)) tuned to one of the four appearing energy gaps $\omega_s$ of the doubly dressed states. In these energy gaps an effective population transfer can occur between the states of the robust TLS, evidenced by signal induced Rabi oscillations at a rate $g'' = g'/2 = g/4$ in the double drive case (Supplementary Note 2).

The measurements take place in the laboratory frame, i.e., all three contributions $\Omega_1$, $\Omega_2$, and $g$ to the population dynamics of the TLS will be visible. In order to see solely the effect of $g$ on the TLS, we alter the modulation of the second drive in Eq. (2) to $\cos(\Omega_1 t + \pi/2)$. This does not change the performance of the scheme, but only changes the axis of rotation to $\sigma_z$ for the second drive $\Omega_2$. Since the readout laser is effectively projecting the population in the $\sigma_z$ eigenbasis, we can make the Rabi rotations of $\Omega_2$ invisible to the readout. To remove the effect of $\Omega_1$ in the data, we can simply sample the measurement at multiple times of $\tau_{\Omega_1} = 2\pi/\Omega_1$, i.e., we measure at times $t = N\tau_{\Omega_1}$ ($N \in \mathbb{N}$). This procedure reveals directly $g''(g')$ as the signal induces Rabi oscillations of the robust qubit under double (single) drive (Fig. 2). Alternatively, we could have applied at the end of the drive a correction pulse in order to complete the full $\Omega_1$ and $\Omega_2$ rotations, so that just the effect of $g''$ remains.

Without a signal, $g$, we achieve coherence times of $T_2^{\Omega_1} \approx 60\,\mu s$ with a single drive ($\Omega_2 = 0$, compare Supplementary Fig. 3 in Supplementary Note 6A) and $T_2^{\Omega_1,\Omega_2} \approx 393\,\mu s$ with a double drive (compare Supplementary Fig. 4 in Supplementary Note 6B). The results for long and slow Rabi oscillations induced by an external signal, $g$, under single and double drive ($\Omega_2/\Omega_1 \approx 0.15$) are shown in Fig. 2. These illustrate a significant increase of the coherence time of the sensor by two orders of magnitude, from $T_2^{\Omega_1} \approx 60\,\mu s$ to a lifetime limited coherence time of $(T_1/2\approx)$ $T_2^{\Omega_1,\Omega_2,g} \approx 1.43\,ms$. It should be noted that the signal itself can be considered as an additional drive (cf. Eq. (3)), correcting external errors $\delta\Omega$ of the previous drive and thereby prolonging the coherence time even further. Consequently, we can improve the bandwidth for high-frequency sensing by almost three orders of magnitude from ~900 kHz (for $T_2^* \approx 1.1\,\mu s$) to ~700 Hz (for a $T_2^{\Omega_1,\Omega_2,g} \approx 1.43\,ms$). Moreover, in Supplementary Note 3 we discuss an improved version of our scheme which has the potential to push the coherence time of the sensor further towards the lifetime limit.

To benchmark the double drive scheme against a standard single drive approach, we determine the smallest magnetic field which can be sensed after an accumulation time $t$. The smallest measurable signal $S$ is eventually bounded by the smallest measurable magnetic field change $\delta B_{\min}$, which is found to be

$$\delta B_{\min}(t, \tau) = \frac{\delta S}{\max\left|\frac{\partial S}{\partial B}\right|} = \frac{1}{\gamma_{NV}} \frac{\sigma(t)}{\alpha \tau C}. \tag{5}$$

Here, $\gamma_{NV}/2\pi = 28.8\,GHz\,T^{-1}$ is the gyromagnetic ratio of the NV defect, $\sigma(t)$ is the standard deviation of the measured normalized fluorescence counts after time $t$, $\alpha$ accounts for a different phase accumulation rate depending on the decoupling scheme, and $C$ is the contrast of the signal (see Supplementary Note 5 for detailed derivation). Since the photon counting is shot noise limited, we have $\sigma(t) = 1/\sqrt{N_{ph} \cdot N}$, with $N_{ph}$ being the number of photons measured in $\tau$ and $N = t/\tau$ is the number of sequence repetitions. With this, Eq. (5) will transform in the commonly known form[47, 48] with some measurement dependent constants.

We recorded $\sigma(t)$ as a function of time and use this to determine $\delta B_{\min}$. The results of this measurement for both the single and double drive are summarized in Fig. 3. The sensitivity

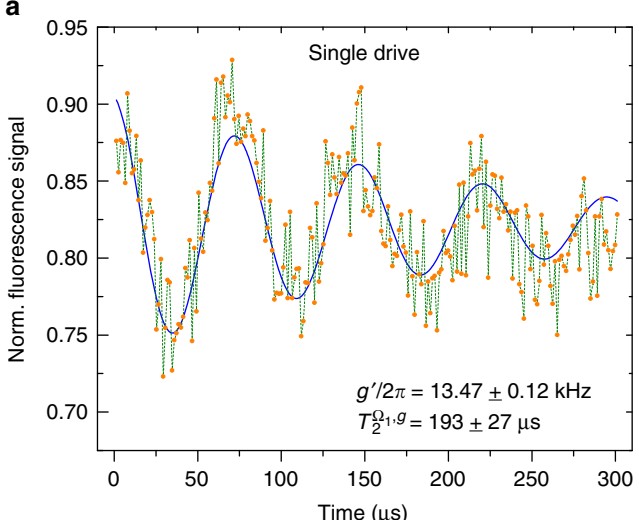

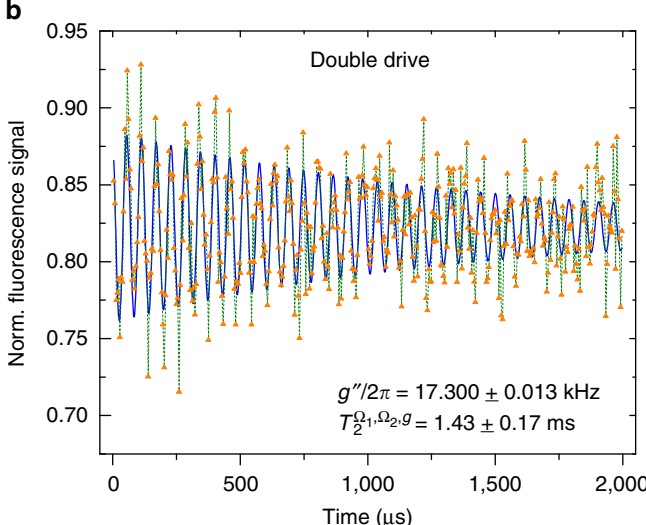

**Fig. 2** Measurements of an external signal of strength $g$. **a** In a single drive approach with $\Omega_1/2\pi = 3.002$ MHz a signal $g' = g/2$ is recorded. **b** By the application of two drive fields with $\Omega_1/2\pi = 3.363$ MHz and $\Omega_2/2\pi = 505$ kHz, we record a signal $g'' = g/4$ and increase the coherence time of the sensor by one order of magnitude with respect to the case of $g = 0$

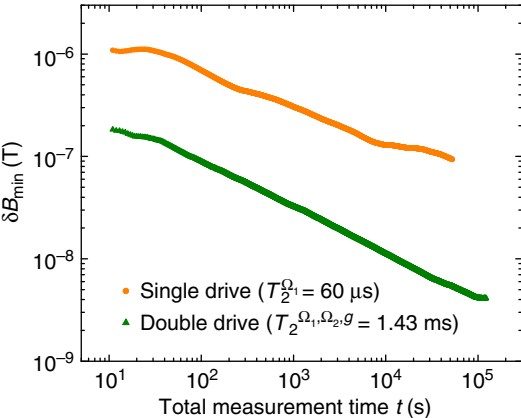

**Fig. 3** Comparison of the smallest measurable magnetic field change $\delta B_{min} = (2\pi \delta g_{min})/\gamma_{NV}$ as a function of total measurement time. To show the total improvement, we obtain $\sigma(t)$ at $\tau = T_2^{\Omega_1} \approx 60$ µs in the single drive case and $\sigma(t)$ at $\tau = T_2^{\Omega_1,\Omega_2,g} \approx 1.43$ ms in the double drive case. Note, that for both data traces a signal was always present, $g/2\pi = 26.9$ kHz and $g/2\pi = 69.2$ kHz in the single and in the double drive, respectively. But only in the double drive the coherence time prolonging effect of $g$ was included into the choice of $\tau$ for Eq. (5) (i.e., the measurement was performed at $\tau = T_2^{\Omega_1,\Omega_2,g}$ instead at $\tau = T_2^{\Omega_1,\Omega_2} \approx 393$ µs)

effect on the coherence time of the sensor (i.e., we measure at $\tau = T_2^{\Omega_1}$ and not at $\tau = T_2^{\Omega_1,g}$). This effect was included in the double drive case.

To examine the signal protection effect more in detail, the coherence time of the sensor is measured as a function of signal strength, $g$, in a single drive configuration (Supplementary Note 7). From these measurements we project the sensitivity associated with a specific signal strength (Fig. 4), assuming $\sigma(t)$ is unchanged for the same repetition $N$. This is a reasonable assumption given that the only difference between measurements is the signal strength, $g$, and sequence length, $\tau$.

Eventually, this phenomenon, which seems to be an inherent part of this continuous scheme, can be used to further increase the performance of the sensor by fine tuning the controlled parameters (static bias field $B_{bias}$ and thereby changing $\omega_0$, $\Omega_1$ and $\Omega_2$) to match the signal frequency, $\omega_s$, and strength, $g$ (see Supplementary Note 3 for further discussions).

## Discussion

We have demonstrated that dynamical decoupling can be used in the context of sensing high frequency fields. In contrast to state-of-the-art pulsed dynamical decoupling protocols, we can show that CDD can be simultaneously integrated into the sensing task. By utilizing a NV center in diamond we have demonstrated by pure concatenation of two drives a coherence time of ~393 µs which constitutes an improvement of more than two orders of magnitude over $T_2^*$, and an increase of resolution from the MHz to a few kHz. The application of this method for wireless communication[49] could have a transformative effect due to the high resolution of the protocol. Since the protocol is applicable to a variety of solid-state, molecular, and atomic systems, we believe that it has a great potential to have a significant impact on many fields and tasks that involve high frequency sensing (up to frequencies in the THz range). Eventually, this method could also be used to improve the coupling to quantum systems[30]. We would like to note that during the preparation of this manuscript we became aware of a related independent work by Joas et al.[50].

can be obtained by $\eta(\tau) = \delta B_{min}(t, \tau)\sqrt{t}$, which is optimal in the vicinity of the coherence time of the sensor, $\tau \approx T_2$. With our system, we achieve a sensitivity of $\eta_{\Omega_1,\Omega_2,g} \lesssim 1$ µT Hz$^{-0.5}$ in the double drive case at ~1.6 GHz, which should be compared to $\eta_{\Omega_1,g} \lesssim 20$ µT Hz$^{-0.5}$ for a single drive approach.

Both traces in Fig. 3 were recorded while a signal $g$ was applied. Apart from the mere fact, that the number of driving fields are different, the specific choice for $\tau$ will also determine the magnitude of the smallest measurable magnetic field change $\delta B_{min}$. Obtaining the coherence time without a signal, $g$, (which are $T_2^{\Omega_1}$ and $T_2^{\Omega_1,\Omega_2}$), is a common practice in the field, but will not result in a correct choice of $\tau$ for the sensitivity measurement and also for Eq. (5), since the signal has an impact on the sensor's sensitivity. However, if $\delta B_{min}$ shall be evaluated correctly, then the non-linearity, i.e., the coherence time prolonging effect of the signal, has to be taken into account. Otherwise an even worse $\delta B_{min}$ will be measured as it is exemplarily shown for the single drive case in Fig. 3, where $\delta B_{min}$ was evaluated and measured under the naive assumption that the signal has no

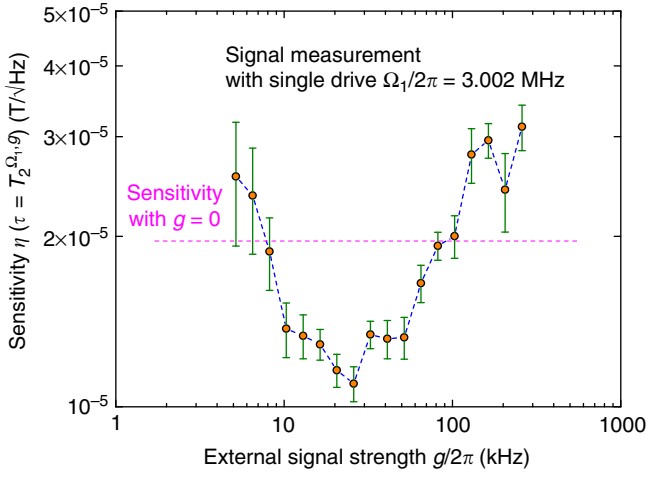

**Fig. 4** Projected sensitivity of the single drive scheme as a function of signal strength, $g$. The coherence time measurements, $T_2^{\Omega_1, g}$ for a fixed drive $\Omega_1$ and subsequently increasing signal strength, $g$, (displayed in Supplementary Fig. 5) are expressed in terms of sensitivity. The magenta dashed line indicates the sensitivity of the sensor if no signal is applied. The figure illustrates that an external signal has a non-linear effect on the sensitivity of the sensor, which has to be taken into account in the sensitivity estimation. The error bars $\Delta \eta \left( T_2^{\Omega_1, g} \right)$ in this graph represent the converted standard deviation $\Delta T_2^{\Omega_1, g}$ from Supplementary Fig. 5

**Data availability**. The authors declare that all relevant data supporting the findings of this study are available within the paper (and its Supplementary Information file). Any raw data can be obtained from the corresponding authors on reasonable request.

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

## Acknowledgements

The experiments presented here were supported by the Qudi Software Suite[51]. A.S., A.H. and U.L.A. acknowledge funding from the Innovation Foundation Denmark through the project EXMAD and the Qubiz center and the Danish Research Council via the Sapere Aude project (DIMS). T.U. and F.J. acknowledge the Volkswagenstiftung. A.R. acknowledges the support of the Israel Science Foundation (grant no. 1500/13), the support of the European commission, EU Project DIADEMS. This project has received funding from the European Union's Horizon 2020 research and innovation program under grant agreement No 667192 Hyperdiamond and Research Cooperation Program and DIP program (FO 703/2-1).

## Author contributions

N.A. and A.R. conceived the idea and developed the theory. A.H., A.R., U.L.A. and F.J. designed and supervised the project. A.S., T.U. and D.L. performed, planned and developed the concept of the experiment. A.S., N.A., T.U. and D.L. analyzed the data. N.A. and D.L. planned and carried out the simulations. A.S. and N.A. took the lead in writing the manuscript. All authors contributed to the interpretation of the results, provided critical feedback and helped to shape the research, analysis and manuscript.
