## [Peer Review File · Nature Communications]

Reviewers' comments:

Reviewer #1 (Remarks to the Author):

The manuscript entitled "Narrow-bandwidth sensing of high-frequency fields with continuous dynamical decoupling" by Stark et al. reports experimental demonstration of quantum sensing of weak magnetic fields with a NV center using continuous dynamical decoupling. Using a concatenation of two microwave driving fields, the authors are able to demonstrate a spin decoherence time as long as 1.43 ms, leading to a magnetic sensing sensitivity of 4 nT.

Earlier experimental studies in this area have almost exclusively used pulsed or time-domain dynamical decoupling to mitigate effects of carbon-13 nuclear spin bath. A drawback of the time domain technique is that the decoupling functions only at the frequency of the repetitive pi-pulses that induce the decoupling. In comparison, the continuous dynamical decoupling (which is based on dressed spin states or spin locking) works as the name implies, i.e. continuously, or as long as the decoupling fields are on. In this regard, the results presented in this manuscript represent a major and remarkable progress in the practical use of NV centers for magnetic sensing. The manuscript is nicely written and the high quality experimental results are clearly presented. I recommend the publication of the manuscript in Nature Communications.

Minor correction:

In Fig. 2b, the spin decoherence time given is 1.43 microsecond. I suspect this is a typo. It should be 1.43 ms.

Reviewer #2 (Remarks to the Author):

The present report discusses the work entitled 'Narrow-bandwidth sensing of high-frequency fields with continuous dynamical decoupling' by A. Stark et al., to be considered for publication in Nature Communications. The paper discusses sensing of high-frequency signals via a two-level qubit and proposes a protocol to go past the usual T_2^* limitation of relaxometry measurements. The protocol is based on protection from environmental noise by operating in a double-rotating frame, where the longitudinal fluctuations correspond to the ones of the driving field and are therefore considerably suppressed with respect to the case of no driving.

It should be stressed here that such protocols (called concatenated continuous dynamical decoupling, or CCD) are not new and have been discussed at length in previous publications (most notably New J. Phys. 14, 093030 (2012), which is Ref. 28 of the present manuscript). In particular, out of the infinite number of driving fields that can be theoretically employed for the case of CCD, in Ref. 28 the use of two driving fields was experimentally investigated, as done in the manuscript under consideration. Eventually, the "high-frequency sensing" discussed in the paper by Stark et al. is a CCD scheme with 3 driving fields. I am therefore rather skeptical regarding the novelty of the work, in particular regarding its publication into Nature Communications. My doubts are further strengthened by the fact that the manuscript does not try to apply the protocol to a "real-world" sensing problem, restricting instead the demonstration of its enhanced sensitivity to the case of an artificial signal.

In addition, I have a few remarks about the content of the paper itself:

- 1) - I have found several confusing passages and notation in the text. For instance:
 - TLS (which I believe stands for two-level system) is not explicitly defined anywhere in the text.
 - In the second column of page 3 the authors discuss the enhancement of the sensor's coherence time from 60 to 400 us in case of a single/double drive and they reference Fig. 2. In Fig. 2, the case of a single/double Rabi drive is shown, with coherence times of 193 us and 1.43 us (!). The

"us" label in panel (b) seems incorrect. I also assume that the difference between 60-400 and 193-1430 is in the fact that the signal "g" is applied in Fig. 2. If so, what happens in Fig. 3? what do these coherence times represent? can the authors clarify this.

- Page 3, first column. I see a drive of the form $\text{Cos}(\Omega_1 + \pi/2)$. I believe time is missing here.

2) - It seems to me that the readout process may have been simplified (i.e. no lab-frame oscillations) by preparing the qubit each time along the quantization axis of the rotating frame with a $\pi/2$ -pulse, before starting the continuous driving. One could then apply the same $\pi/2$ -pulses before readout. In the case of a single drive, this is routinely done for the measurement of T_1 -rho (see e.g. the T_1 -rho scheme at the top of Fig.2 of PRL 112, 147602 - 2014). Could the authors comment on this?

In summary, the paper could be interesting to the community if it tried to include further experimental work or protocols, e.g. beyond the mere testing of the sensing protocol published in New J. Phys. 14, 093030 (2012) with an artificial signal. In its present form, the manuscript in my opinion shows too little novelty and a too strong overlap with previous work to justify its publication in Nature Communications.

Reviewer #3 (Remarks to the Author):

In their manuscript entitled "Narrow-bandwidth sensing of high-frequency fields with continuous dynamical decoupling", the authors demonstrate the high-sensitivity, gigahertz magnetic field sensing by utilizing a concatenated continuous dynamical decoupling technique and an electronic spin of a single nitrogen-vacancy center in diamond. The doubly-dressed states created by this technique show the coherence time up to 1.43 ms and are capable of detecting the 4-nT magnetic fields at 1.6 GHz.

While the research topic is surely of importance and the experimental data are overall of high quality, I find the presentation unsatisfactory. A revision must be made before I can make a judgement on whether the present work is worthy of publication in Nature Communications. See my comments below.

The first three comments are primarily about the notations. I strongly feel that the present authors should be more careful about them to enhance the readability of the manuscript.

(1) In the main text, there are seven "see supplementary" instructions without mentioning which part of the relatively long 14-page supplementary to look at. This is very unkind and even irritating, especially when the contents of the supplementary are not ordered in accordance with the main text. For instance, in the second "see supplementary" instruction about analog pulsed version of the scheme (page 2), one has to look at the very last section of the supplementary (Sec. IX, page 13). The instructions should say "see supplementary Sec. xx". In addition, the equations and figures in the supplementary are confusingly numbered 1, 2, 3... as in the main text. They should be labeled as S1, S2, S3... for instance. In the following, I will use these labels to avoid confusion.

(2) The distinction between the two terms "coherence time" and "lifetime" is unclear. In page 1 of the main text, "lifetime" means T_1 , while in the supplementary the quantities that were called "coherence times" in the main text are called "lifetimes". It may be that the distinction is subtle in dealing with the dressed states, but still, the interchangeable use of them should be avoided. Moreover, the term "the lifetime of the signal" appearing both in the main text and in the supplementary is incomprehensible to me. The coherence time (or the lifetime as the authors say) measured under non-zero g is still that of the sensor with the signal effectively working as the

third drive (say, the triply-dressed state). On the other hand, the signal here is externally applied and sustained during the measurement. Why do the authors use the term "the lifetime of the signal"?

(3) There are four important timescales extracted in the present work: (i) the coherence time under single drive with $g = 0$, (ii) the coherence time under single drive with nonzero g , (iii) the coherence time under double drive with $g = 0$, (iv) the coherence time under double drive with nonzero g . In addition to the mixed use of "coherence time" and "lifetime" as mentioned above, the notations to these timescales are not consistent. The authors interchangeably use " T_{coh} " (Figs. 2, 3), " $T_{\{\Omega_1/\Omega_1, \Omega_2\}}$ " (page 3 of the main text), " T_2 " (Sec. VIII of the supplementary).

I suggest that they be given separate and systematic notations to avoid confusions. It would be unmistakable if, for instance, they are explicitly defined as $T_{\{\Omega_1\}}$ for (i), $T_{\{\Omega_1, g\}}$ for (ii), $T_{\{\Omega_1, \Omega_2\}}$ for (iii), and $T_{\{\Omega_1, \Omega_2, g\}}$ for (iv), and denoted so consistently throughout the main text and the supplementary. In the following, I will use these notations to avoid confusion.

Below are the comments on the scientific contents of the work.

(4) From the manuscript, it is difficult to judge what is conceptually new in the present work. In particular, the statement in page 1, "we propose and experimentally demonstrate a sensing scheme that is capable of probing high frequency signals with unprecedented sensitivity", is vague. The central idea of the present work, concatenated continuous dynamical decoupling, has already been discussed in detail in Ref. 34. It seems to me that the only ingredient added in the present work is the observation that the signal phase can be unknown. Indeed, apart from the sensitivity, the single drive is sufficient to demonstrate the GHz sensing (Fig. 3). In other words, if one regards the second drive in Ref. 34 as a signal, the GHz sensing has already been achieved (in my understanding, the second drive used in Ref. 34 is not phase-matched to the first drive, so it can be regarded as a signal). The manuscript should be revised to clearly separate what is really new in the present work from what has already been known or demonstrated.

(5) The authors claim "that by using a concatenation of two drives, an improvement in coherence time of the sensor by more than one order of magnitude is achieved" (page 2, right). By this, I presume that the authors refer to the results that $T_{\{\Omega_1\}} = 40 \mu\text{s}$ for $\Omega_1 = 3.363 \text{ MHz}$ (Fig. S3) and $T_{\{\Omega_1, \Omega_2\}} = 400 \mu\text{s}$ for $\Omega_1 = 3.363 \text{ MHz}$ and $\Omega_2 = 505 \text{ kHz}$ (Fig. S4). It may be that the authors also include $T_{\{\Omega_1, \Omega_2, g\}} = 1.43 \text{ ms}$ for $\Omega_1 = 3.363 \text{ MHz}$, $\Omega_2 = 505 \text{ kHz}$, and $g = g/4 = 17.3 \text{ kHz}$ as the improvement. However, this is not easy to recognize, because in the main text the results for different conditions are discussed ($T_{\{\Omega_1\}} = 60 \mu\text{s}$ for $\Omega_1 = 3.002 \text{ MHz}$ and $T_{\{\Omega_1, \Omega_2\}} = 400 \mu\text{s}$).

Moreover, in Ref. 34, the extension of the coherence time from $2.3 \mu\text{s}$ (single drive) to $21 \mu\text{s}$ (double drive) has been demonstrated. While the present work has achieved longer timescale than did in Ref. 34, the degree of the extension is almost the same.

The authors should explain what is really new about the improved coherence time in the present work, and present experimental data (preferably in the main text) to fully justify the claim.

(6) Even though the title claims "narrow-bandwidth", the authors have made no mention of the bandwidth of the protocol in the main text or the supplementary. The discussion on the attainable and achieved bandwidths, based on both theory and experiments, must be provided.

(7) In Fig. 3, $T_{\text{coh}} = 60 \mu\text{s}$ for single drive and $T_{\text{coh}} = 1.43 \text{ ms}$ for double drive are stated. Based on the numbers given, I presume that they mean $T_{\{\Omega_1\}} = 60 \mu\text{s}$ and $T_{\{\Omega_1, \Omega_2, g\}} = 1.43 \text{ ms}$, respectively. On the other hand, according to Sec. V of the supplementary, Fig. 3 is obtained by the slope measurements as varying g . Then, it seems appropriate to compare $T_{\{\Omega_1, g\}} = 193 \mu\text{s}$ and $T_{\{\Omega_1, \Omega_2, g\}} = 1.43 \text{ ms}$ (even more confusingly, the caption says " $T_{\text{coh}} = 60 \mu\text{s}$ in the signal drive case"). Why do the authors try to compare between $T_{\{\Omega_1\}} = 60 \mu\text{s}$ and $T_{\{\Omega_1, \Omega_2, g\}} = 1.43 \text{ ms}$ in Fig. 3?

(8) In Fig. S5, the dashed horizontal line is drawn at $T_{\{\Omega_1, g\}}$ (label as "lifetime of signal g") $\approx 50 \mu\text{s}$ and denoted as "lifetime without g". Where does this value come from? In Fig. S4, $T_{\{\Omega_1\}}$ (labeled as "lifetime of Ω_1 ") is $\approx 60 \mu\text{s}$ at $\Omega_1 = 3.002 \text{ MHz}$, which is consistent with the main text and should appear in Fig. S5. As the dashed horizontal line in Fig. S5 is translated into that in Fig. 4 (indicated as "sensitivity with $g = 0$ "), the reason for using $50 \mu\text{s}$ rather than the naively expected $60 \mu\text{s}$ should be explained.

Finally, below is a list of minor problems/typographical errors I have noticed. They should be corrected as well.

(9) In the main text, the term "double dressed state" is used, while "double dressed state", "double-dressed state", and "doubly-dressed state" appear randomly in the supplementary. Unify the expression. I feel "doubly-dressed state" most accurate grammatically.

(10) In page 1, left column, line 2, "of a great significance" should read "of great significance".

(11) In page 1, left column, line 4, the defined abbreviation "NMR" has never been used in the rest of the manuscript. It can be left out.

(12) In page 1, left column, line 31, the abbreviation "TLS" is used without definition.

(13) In page 1, right column, line 2, "NV" should read "nitrogen-vacancy (NV)", as the abbreviation "NV" is used for first time in the main text. On the other hand, within the abstract, "NV" is defined but never used, so can be left out.

(14) In Fig. 2(b), " $1.43 \pm 0.17 \mu\text{s}$ " in the figure should read " $1.43 \pm 0.17 \text{ ms}$ ".

(15) In Ref. 36, "arXiv:1611.01515" has been published in "Journal of Optics 19, 044003 (2017)".

(16) In Ref. 40, the publication year (2004) is missing.

(17) In Ref. 43, the journal page should be "227" or "227–234".

(18) In Fig. S4, the label for the horizontal axis should read " Ω_2/Ω_1 " (" Ω_1/Ω_2 " is larger than 1).

(19) In page 10 of the supplementary, "Furthermore, it seems that Ω_2 introduces more noise to the system as g does." should read "Furthermore, it seems that Ω_2 introduces more noise to the system than g does."

Complete comment of referee 1

The manuscript entitled "Narrow-bandwidth sensing of high-frequency fields with continuous dynamical decoupling" by Stark et al. reports experimental demonstration of quantum sensing of weak magnetic fields with a NV center using continuous dynamical decoupling. Using a concatenation of two microwave driving fields, the authors are able to demonstrate a spin decoherence time as long as 1.43 ms, leading to a magnetic sensing sensitivity of 4 nT.

Earlier experimental studies in this area have almost exclusively used pulsed or time-domain dynamical decoupling to mitigate effects of carbon-13 nuclear spin bath. A drawback of the time domain technique is that the decoupling functions only at the frequency of the repetitive pi-pulses that induce the decoupling. In comparison, the continuous dynamical decoupling (which is based on dressed spin states or spin locking) works as the name implies, i.e. continuously, or as long as the decoupling fields are on. In this regard, the results presented in this manuscript represent a major and remarkable progress in the practical use of NV centers for magnetic sensing. The manuscript is nicely written and the high quality experimental results are clearly presented. I recommend the publication of the manuscript in Nature Communications.

Minor correction: In Fig. 2b, the spin decoherence time given is 1.43 microsecond. I suspect this is a typo. It should be 1.43 ms.

Reply to referee 1

In the following, we want to give a detailed reply on each comment of the referee. For this purpose, we will label each comment.

Comment 1.1: *"Earlier experimental studies in this area have almost exclusively used pulsed or time-domain dynamical decoupling to mitigate effects of carbon-13 nuclear spin bath. A drawback of the time domain technique is that the decoupling functions only at the frequency of the repetitive pi-pulses that induce the decoupling. In comparison, the continuous dynamical decoupling (which is based on dressed spin states or spin locking) works as the name implies, i.e. continuously, or as long as the decoupling fields are on. In this regard, the results presented in this manuscript represent a major and remarkable progress in the practical use of NV centers for magnetic sensing. The manuscript is nicely written and the high quality experimental results are clearly presented. I recommend the publication of the manuscript in Nature Communications."*

Response: We thank the referee for his/her very positive evaluation of our manuscript and recommendation for publication.

Comment 1.2: *"In Fig. 2b, the spin decoherence time given is 1.43 microsecond. I suspect this is a typo. It should be 1.43 ms."*

Response: That was indeed a typo and the unit 'ms' should have been used.

Changes made: In Fig. 2b the unit of the spin coherence time has been corrected.

Complete comment of referee 2

The present report discusses the work entitled 'Narrow-bandwidth sensing of high-frequency fields with continuous dynamical decoupling' by A. Stark et al., to be considered for publication in Nature Communications. The paper discusses sensing of high-frequency signals via a two-level qubit and proposes a protocol to go past the usual $T2^*$ limitation of relaxometry measurements. The protocol is based on protection from environmental noise by operating in a double-rotating frame, where the longitudinal fluctuations correspond to the ones of the driving field and are therefore considerably suppressed with respect to the case of no driving.

It should be stressed here that such protocols (called concatenated continuous dynamical decoupling, or CCD) are not new and have been discussed at length in previous publications (most notably New J. Phys. 14, 093030 (2012), which is Ref. 28 of the present manuscript). In particular, out of the infinite number of driving fields that can be theoretically employed for the case of CCD, in Ref. 28 the use of two driving fields was experimentally investigated, as done in the manuscript under consideration. Eventually, the "high-frequency sensing" discussed in the paper by Stark et al. is a CCD scheme with 3 driving fields. I am therefore rather skeptical regarding the novelty of the work, in particular regarding its publication into Nature Communications. My doubts are further strengthened by the fact that the manuscript does not try to apply the protocol to a "real-world" sensing problem, restricting instead the demonstration of its enhanced sensitivity to the case of an artificial signal.

In addition, I have a few remarks about the content of the paper itself:

1) - I have found several confusing passages and notation in the text. For instance: - TLS (which I believe stands for two-level system) is not explicitly defined anywhere in the text. - In the second column of page 3 the authors discuss the enhancement of the sensor's coherence time from 60 to 400 us in case of a single/double drive and they reference Fig. 2. In Fig. 2, the case of a single/double Rabi drive is shown, with coherence times of 193 us and 1.43 us (!). The "us" label in panel (b) seems incorrect. I also assume that the difference between 60-400 and 193-1430 is in the fact that the signal "g" is applied in Fig. 2. If so, what happens in Fig. 3? what do these coherence times represent? can the authors clarify this. - Page 3, first column. I see a drive of the form $\cos(\Omega_1 + \pi/2)$. I believe time is missing here.

2) - It seems to me that the readout process may have been simplified (i.e. no lab-frame oscillations) by preparing the qubit each time along the quantization axis of the rotating frame with a $\pi/2$ -pulse, before starting the continuous driving. One could then apply the same $\pi/2$ -pulses before readout. In the case of a single drive, this is routinely done for the measurement of $T1$ -rho (see e.g. the $T1$ -rho scheme at the top of Fig.2 of PRL 112, 147602 - 2014). Could the authors comment on this?

In summary, the paper could be interesting to the community if it tried to include further experimental work or protocols, e.g. beyond the mere testing of the sensing protocol published in New J. Phys. 14, 093030 (2012) with an artificial signal. In its present form, the manuscript in my opinion shows too little novelty and a too strong overlap with previous work to justify its publication in Nature Communications.

Reply to referee 2

We appreciate the critical review of the referee and we would like to thank him/her for pointing out a few aspects to be improved, in particular with respect to the novelty of our work. Below we provide a detailed reply on each comment indicating also the changes made in the revised manuscript.

Comment 2.1: *"It should be stressed here that such protocols (called concatenated conti-*

nuous dynamical decoupling, or CCD) are not new and have been discussed at length in previous publications (most notably *New J. Phys.* 14, 093030 (2012), which is Ref. 28 of the present manuscript). In particular, out of the infinite number of driving fields that can be theoretically employed for the case of CCD, in Ref. 28 the use of two driving fields was experimentally investigated, as done in the manuscript under consideration. Eventually, the "high-frequency sensing" discussed in the paper by Stark et al. is a CCD scheme with 3 driving fields. I am therefore rather skeptical regarding the novelty of the work, in particular regarding its publication into *Nature Communications*. My doubts are further strengthened by the fact that the manuscript does not try to apply the protocol to a "real-world" sensing problem, restricting instead the demonstration of its enhanced sensitivity to the case of an artificial signal."

Response: Given the content of the comment we assume that the given reference (New J. Phys. 14, 093030 (2012)) was mistaken. Instead we think that the referee is referring to New Journal of Physics 14, 113023 (2012), which is Ref. 36 in the revised manuscript.

It is correct that Ref. 36 shows a detailed analysis of concatenated continuous dynamical decoupling (CCDD) protocols, but it should be stressed that these investigations were performed in the context of protecting the qubit from the drive and environmental noise under the assumption of phase-matched driving fields. Neither the integration nor the investigation of CCDD in a sensing tasks were carried out in Ref. 36, which is the central aspect of the present manuscript.

Moreover, the construction of the second drive in Ref. 36 is different in comparison to the present work and is the reason for a worse performance of the decoupling procedure. In Ref. 36, the second field drives only one of the appearing dressed state transitions instead of both (see Sec. IV in the supplementary of our manuscript), which is also stated by the authors in Ref. 36 in the middle of the last paragraph before Sec. 4: "[...] *In the experiment we used the simplified second-order field $H_{d2} = \hbar\Omega_2 \cos((\omega + \Omega_1)t + \phi)\sigma_x$ for which some counter-rotating terms persist that limit the achievable coherence time. [...]*"

Besides the fact that sensing was not demonstrated previously with concatenated CDD it was also unclear whether it is possible to simultaneously sense and protect the qubit for fields in the GHz range. Although our artificial signal can be comprised as a third drive, we analyze and show that the signal phase does not have to be matched to the phase of the protection scheme. To the best of our knowledge, we demonstrate for the first time a narrow-bandwidth sensing of high-frequency fields.

A more detailed elaboration in terms of novelty of our contribution is given in the response to Comment 2.7.

Changes made: We agree with the referee that the separation of the present work from previous works was not clear enough and hence we changed the following part in the manuscript. In the main text, third paragraph on page 1 after the first sentence:

"However, the significance of CDD for sensing high frequency

fields remained elusive. Indeed, it was unclear whether it is possible to incorporate such a protection into the metrology task of sensing frequencies in the GHz domain. The first step towards this goal was done recently by integrating CDD in the sensing of high frequency fields with three level systems [39]”

At the beginning of the part ‘The Sensing Scheme’ we include after the first sentence:

“The concatenation of two phase-matched driving fields results in a robust qubit [34]. In what follows we show that such a robust qubit can be utilized as a sensor for frequencies in the range of the qubit’s energy separation and hence, dynamical decoupling can be integrated into the sensing task.”

Comment 2.2: *“TLS (which I believe stands for two-level system) is not explicitly defined anywhere in the text.”*

Response: Indeed, the abbreviation was not introduced in the previous version of the manuscript.

Changes made: In the fourth paragraph on page 1, the second sentence contains the explanation for the abbreviation:

“Unlike relaxation measurements comprising a bandwidth $\propto 1/T_2^*$, determined by the pure dephasing time, T_2^* , of the sensor (up to the MHz range), our protocol overcomes the imposed limitation by protecting the addressed two-level system (TLS) with an adapted concatenated CDD approach.”

Comment 2.3: *“In the second column of page 3 the authors discuss the enhancement of the sensor’s coherence time from 60 to 400 us in case of a single/double drive and they reference Fig. 2. In Fig. 2, the case of a single/double Rabi drive is shown, with coherence times of 193 us and 1.43 us (!). The ”us” label in panel (b) seems incorrect.”*

Response: We changed the labelling of the coherence times in order to indicate under which conditions the respective coherence times were recorded. We also include the reference to the measurements of $T_2^{\Omega_1} \approx 60 \mu\text{s}$ and $T_2^{\Omega_1, \Omega_2} \approx 393 \mu\text{s}$ in the supplementary Sec VI.A and Sec. VI.B, respectively, so that it is not confused with the measurements in the main text in Fig.2.

The label in Fig. 2 was corrected in Comment 1.2.

Changes made: On page 3, starting with the last sentence in the left column which reads now

“Without a signal, g , we achieve coherence times of $T_2^{\Omega_1} \approx 60 \mu\text{s}$ with a single drive ($\Omega_2 = 0$, compare Fig. S3 in Sec. VI.A in Supplementary) and $T_2^{\Omega_1, \Omega_2} \approx 393 \mu\text{s}$ with a double drive (compare Fig. S4 in Sec. VI.B in Supplementary). The results for long and slow Rabi oscillations induced by an external signal, g , under single and double drive ($\Omega_2/\Omega_1 \approx 0.15$) are shown in Fig. 2.”

Comment 2.4: *“I also assume that the difference between 60-400 and 193-1430 is in the fact that the signal ”g” is applied in Fig. 2. If so, what happens in Fig. 3? what do these coherence times represent? can the authors clarify this.”*

Response: Indeed, there is a considerable difference in coherence time of the sensor, if it is measured with a signal, g , and without it. By introducing a consistent labelling of the coherence times (see also Comment 3.3 and Comment 3.9) we clarify under which conditions the measurements have been performed.

Fig. 3 in the main text shows the smallest achievable sensitivity, η_{\min} , and the smallest measurable magnetic field, δB_{\min} , which can be measured using this sensor either with an unoptimized single drive approach and an optimized double drive configuration. Here the term ‘optimized’ refers to the correct choice of the measurement time τ , taking into account the impact of the signal on the coherence time of the sensor (see equation (5) in the main text). If the impact of the signal on the sensor is not taken into account, then we will not obtain the smallest sensitivity with this scheme as denoted in equation (5).

Changes made: We considerably modified the caption of Fig. 3 to

“To show the total improvement, we obtain $\sigma(t)$ at $\tau = T_2^{\Omega_1} \approx 60 \mu\text{s}$ in the single drive case and $\sigma(t)$ at $\tau = T_2^{\Omega_1, \Omega_2, g} \approx 1.43 \text{ ms}$ in the double drive case. Note, that for both data traces a signal was always present, $g = 26.9 \text{ kHz}$ and $g = 69.2 \text{ kHz}$ in the single and in the double drive, respectively. But only in the double drive the coherence time prolonging effect of g was included into the choice of τ for equation (5) (i.e. the measurement was performed at $\tau = T_2^{\Omega_1, \Omega_2, g}$ instead at $\tau = T_2^{\Omega_1, \Omega_2} \approx 393 \mu\text{s}$).”

Moreover, we add a more detailed elaboration on Fig. 3 in the ‘Results’ section of the main text,

“Both traces in Fig. 3 were recorded while a signal g was applied. Apart from the mere fact, that the number of driving fields are different, the specific choice for τ will also determine the magnitude of the smallest measurable magnetic field change δB_{\min} . Obtaining the coherence time without a signal, g , (which are $T_2^{\Omega_1}$ and $T_2^{\Omega_1, \Omega_2}$), is a common practice in the field, but will not result in a correct choice of τ for the sensitivity measurement and also for equation (5), since the signal has an impact on the sensor’s sensitivity. However, if δB_{\min} shall be evaluated correctly, then the non-linearity, i.e. the coherence time prolonging effect of the signal, has to be taken into account. Otherwise an even worse δB_{\min} will be measured as it is exemplarily shown for the single drive case in Fig. 3, where δB_{\min} was evaluated and measured under the naive assumption that the signal has no effect on the coherence time of the sensor (i.e. we measure at $\tau = T_2^{\Omega_1}$ and not at $\tau = T_2^{\Omega_1, g}$). This effect was included in the double drive case.”

Comment 2.5: *“Page 3, first column. I see a drive of the form $\cos(\Omega_1 + \pi/2)$. I believe time is missing here.”*

Response: We thank the referee for pointing out this mistake.

Changes made: We include the missing parameter t and the sentence reads now:

“In order to see solely the effect of g on the TLS, we alter the modulation of the second drive in equation (1) to $\cos(\Omega_1 t + \pi/2)$.”

Comment 2.6: *“It seems to me that the readout process may have been simplified (i.e. no lab-frame oscillations) by preparing the qubit each time along the quantization axis of the rotating frame with a $\pi/2$ -pulse, before starting the continuous driving. One could then apply the same $\pi/2$ -pulses before readout. In the case of a single drive, this is routinely done for the measurement of $T1$ -rho (see e.g. the $T1$ -rho scheme at the top of Fig.2 of PRL 112, 147602 - 2014). Could the authors comment on this?”*

Response: That is correct for the case of a single drive, but for a situation with several drives the incorporation of a $\pi/2$ pulse (under the consideration of all the drives) is not straight forward any longer and becomes more complicated. If we would introduce the $\pi/2$ pulse in the second term of the second drive (see equation (1) on page 2), we in fact perform something similar, i.e. moving into the rotating frame of the modulation of the second drive. But then, we will always see the contributions of Ω_1 . We could also select the rotating frame of Ω_1 , thus, solely the Ω_2 rotations would become visible. Since both drives are perpendicular to each other, a contribution of Ω_1 or Ω_2 will always be present (or even both in other cases) when introducing a $\pi/2$ pulse.

For this reason we decided to sample the measurement as described in the main text (Sec. 'Results' in the second paragraph).

Comment 2.7: *“In summary, the paper could be interesting to the community if it tried to include further experimental work or protocols, e.g. beyond the mere testing of the sensing protocol published in New J. Phys. 14, 093030 (2012) with an artificial signal. In its present form, the manuscript in my opinion shows too little novelty and a too strong overlap with previous work to justify its publication in *Nature Communications*.”*

Response: Concerning the relation of the mentioned article (New J. Phys. 14, 093030 (2012)) to the present manuscript, we would like to refer to the discussion in Comment 2.1, which states that in (New J. Phys. 14, 093030 (2012)) concatenated continuous dynamical decoupling (CCDD) was solely used for protecting the qubit and was not considered to be integrated in a sensing task.

Besides the sensing application of concatenated CDD, the task of high frequency sensing (in the GHz regime) was only achievable with relaxometry measurements with a T_2^* bandwidth limit. In the present work, we show for the first time that high frequency sensing is possible in a continuous and pulsed fashion (the pulsed proposal is in the supplementary Sec. IX), which results in a largely improved bandwidth limited by the T_2 time of the sensor. Our work also demonstrates that the metrology task of high frequency sensing is no longer only possible with relaxometry measurements.

We would like to emphasize that the present approach (which utilized an adjusted version of concatenated CDD) is not only limited to cases where 'high frequency' refers to GHz fields. Instead, the approach can be applied to any two level system and hence also frequencies in the THz range can be sensed considering physical systems with transitions in this frequency range.

Because our scheme enables to achieve the enhanced resolution of high frequency sensing (in our case an increased of resolution from the MHz to a few kHz), which has not been achieved before, it has a great potential to have a significant impact on many fields and tasks that involve high frequency sensing, such as wireless communication, coupling to quantum systems, classical field sensing, detection of electron spins in solids, and nuclear magnetic resonance spectroscopy.

In conclusion, we think that with these achievements it justified that our work can be considered as a new contribution to the field, which warrants publication in Nature Communications.

Changes made: To point out our contribution and to separate it from previous work, we modified the third and fourth paragraph in the introduction on the first page, left column to:

“However, the significance of CDD for sensing high frequency fields remained elusive. Indeed, it was unclear whether it is possible to incorporate such a protection into the metrology task of sensing frequencies in the GHz domain. The first step towards this goal was done recently by integrating concatenated CDD in the sensing of high frequency fields with three level systems [39]. In this article, we propose, analyze and experimentally demonstrate for the first time a sensing scheme that is capable of probing high frequency signals with a coherence time, T_2 , limited sensitivity. Unlike relaxation measurements comprising a bandwidth $\propto 1/T_2^*$, determined by the pure dephasing time, T_2^* , of the sensor (up to the MHz range), our protocol overcomes the imposed limitation by protecting the addressed two-level system (TLS) with an adapted concatenated CDD approach. We use and adjust it such that high frequency sensing becomes feasible even for not phase-matched signals. ”

In addition, we point out in the beginning of the Sec. 'The Sensing Scheme' starting from the second sentence the aim of our approach:

“The concatenation of two phase-matched driving fields results in a robust qubit [34]. In what follows we show that such a robust qubit can be utilized as a sensor for frequencies in the range of the qubit's energy separation and hence, dynamical decoupling can be integrated into the sensing task.”

In the main text of the 'Conclusion' section, we add to the end of the paragraph:

“Since the protocol is applicable to a variety of solid-state, molecular, and atomic systems, we believe that it has a great potential to have a significant impact on many fields and tasks that involve high frequency sensing (up to frequencies in the THz range)”

Complete comment of referee 3

In their manuscript entitled "Narrow-bandwidth sensing of high-frequency fields with continuous dynamical decoupling", the authors demonstrate the high-sensitivity, gigahertz magnetic field sensing by utilizing a concatenated continuous dynamical decoupling technique and an electronic spin of a single nitrogen-vacancy center in diamond. The doubly-dressed states created by this technique show the coherence time up to 1.43 ms and are capable of detecting the 4-nT magnetic fields at 1.6 GHz.

While the research topic is surely of importance and the experimental data are overall of high quality, I find the presentation unsatisfactory. A revision must be made before I can make a judgement on whether the present work is worthy of publication in Nature Communications. See my comments below.

The first three comments are primarily about the notations. I strongly feel that the present authors should be more careful about them to enhance the readability of the manuscript.

(1) In the main text, there are seven "see supplementary" instructions without mentioning which part of the relatively long 14-page supplementary to look at. This is very unkind and even irritating, especially when the contents of the supplementary are not ordered in accordance with the main text. For instance, in the second "see supplementary" instruction about analog pulsed version of the scheme (page 2), one has to look at the very last section of the supplementary (Sec. IX, page 13). The instructions should say "see supplementary Sec. xx". In addition, the equations and figures in the supplementary are confusingly numbered 1, 2, 3... as in the main text. They should be labeled as S1, S2, S3... for instance. In the following, I will use these labels to avoid confusion.

(2) The distinction between the two terms "coherence time" and "lifetime" is unclear. In page 1 of the main text, "lifetime" means T_1 , while in the supplementary the quantities that were called "coherence times" in the main text are called "lifetimes". It may be that the distinction is subtle in dealing with the dressed states, but still, the interchangeable use of them should be avoided. Moreover, the term "the lifetime of the signal" appearing both in the main text and in the supplementary is incomprehensible to me. The coherence time (or the lifetime as the authors say) measured under non-zero g is still that of the sensor with the signal effectively working as the third drive (say, the triply-dressed state). On the other hand, the signal here is externally applied and sustained during the measurement. Why do the authors use the term "the lifetime of the signal"?

(3) There are four important timescales extracted in the present work: (i) the coherence time under single drive with $g = 0$, (ii) the coherence time under single drive with nonzero g , (iii) the coherence time under double drive with $g = 0$, (iv) the coherence time under double drive with nonzero g . In addition to the mixed use of "coherence time" and "lifetime" as mentioned above, the notations to these timescales are not consistent. The authors interchangeably use " T_{coh} " (Figs 2, 3), " $T_{\Omega_1/\Omega_1, \Omega_2}$ " (page 3 of the main text), " T_2 " (Sec. VIII of the supplementary).

I suggest that they be given separate and systematic notations to avoid confusions. It would be unmistakable if, for instance, they are explicitly defined as T_{Ω_1} for (i), $T_{\Omega_1, g}$ for (ii), T_{Ω_1, Ω_2} for (iii), and $T_{\Omega_1, \Omega_2, g}$ for (iv), and denoted so consistently throughout the main text and the supplementary. In the following, I will use these notations to avoid confusion.

Below are the comments on the scientific contents of the work.

(4) From the manuscript, it is difficult to judge what is conceptually new in the present work. In particular, the statement in page 1, "we propose and experimentally demonstrate a sensing scheme that is capable of probing high frequency signals with unprecedented sensitivity", is vague. The central idea of the present work, concatenated continuous dynamical decoupling, has already been discussed in detail in Ref. 36. It seems to me that the only ingredient added in the present work is the observation that the signal phase can be unknown. Indeed, apart

from the sensitivity, the single drive is sufficient to demonstrate the GHz sensing (Fig. 3). In other words, if one regards the second drive in Ref. 36 as a signal, the GHz sensing has already been achieved (in my understanding, the second drive used in Ref. 36 is not phase-matched to the first drive, so it can be regarded as a signal). The manuscript should be revised to clearly separate what is really new in the present work from what has already been known or demonstrated.

(5) The authors claim "that by using a concatenation of two drives, an improvement in coherence time of the sensor by more than one order of magnitude is achieved" (page 2, right). By this, I presume that the authors refer to the results that $T_{\Omega_1} = 40 \mu\text{s}$ for $\Omega_1 = 3.363 \text{ MHz}$ (Fig. S3) and $T_{\Omega_1, \Omega_2} = 400 \mu\text{s}$ for $\Omega_1 = 3.363 \text{ MHz}$ and $\Omega_2 = 505 \text{ kHz}$ (Fig. S4). It may be that the authors also include $T_{\Omega_1, \Omega_2, g} = 1.43 \text{ ms}$ for $\Omega_1 = 3.363 \text{ MHz}$, $\Omega_2 = 505 \text{ kHz}$, and $g'' = g/4 = 17.3 \text{ kHz}$ as the improvement. However, this is not easy to recognize, because in the main text the results for different conditions are discussed ($T_{\Omega_1} = 60 \mu\text{s}$ for $\Omega_1 = 3.002 \text{ MHz}$ and $T_{\Omega_1, \Omega_2} = 400 \mu\text{s}$) Moreover, in Ref. 36, the extension of the coherence time from $2.3 \mu\text{s}$ (single drive) to $21 \mu\text{s}$ (double drive) has been demonstrated. While the present work has achieved longer timescale than did in Ref. 36, the degree of the extension is almost the same. The authors should explain what is really new about the improved coherence time in the preset work, and present experimental data (preferably in the main text) to fully justify the claim.

(6) Even though the title claims "narrow-bandwidth", the authors have made no mention of the bandwidth of the protocol in the main text or the supplementary. The discussion on the attainable and achieved bandwidths, based on both theory and experiments, must be provided.

(7) In Fig. 3, $T_{\text{coh}} = 60 \mu\text{s}$ for single drive and $T_{\text{coh}} = 1.43 \text{ ms}$ for double drive are stated. Based on the numbers given, I presume that they mean $T_{\Omega_1} = 60 \mu\text{s}$ and $T_{\Omega_1, \Omega_2, g} = 1.43 \text{ ms}$, respectively. On the other hand, according to Sec. V of the supplementary, Fig. 3 is obtained by the slope measurements as varying g . Then, it seems appropriate to compare $T_{\Omega_1, g} = 193 \mu\text{s}$ and $T_{\Omega_1, \Omega_2, g} = 1.43 \text{ ms}$ (even more confusingly, the caption says " $T_{\text{coh}} = 60 \mu\text{s}$ in the signal drive case"). Why do the authors try to compare between $T_{\Omega_1} = 60 \mu\text{s}$ and $T_{\Omega_1, \Omega_2, g} = 1.43 \text{ ms}$ in Fig. 3?

(8) In Fig. S5, the dashed horizontal line is drawn at $T_{\Omega_1, g}$ (label as "lifetime of signal g ") $\approx 50 \mu\text{s}$ and denoted as "lifetime without g ". Where does this value come from? In Fig. S4, T_{Ω_1} (labeled as "lifetime of Ω_1 ") is $\approx 60 \mu\text{s}$ at $\Omega_1 = 3.002 \text{ MHz}$, which is consistent with the main text and should appear in Fig. S5. As the dashed horizontal line in Fig. S5 is translated into that in Fig. 4 (indicated as "sensitivity with $g = 0$ "), the reason for using $50 \mu\text{s}$ rather than the naively expected $60 \mu\text{s}$ should be explained.

Finally, below is a list of minor problems/typographical errors I have noticed. They should be corrected as well.

(9) In the main text, the term "double dressed state" is used, while "double dressed state", "double-dressed state", and "doubly-dressed state" appear randomly in the supplementary. Unify the expression. I feel "doubly-dressed state" most accurate grammatically.

(10) In page 1, left column, line 2, "of a great significance" should read "of great significance".

(11) In page 1, left column, line 4, the defined abbreviation "NMR" has never been used in the rest of the manuscript. It can be left out.

(12) In page 1, left column, line 31, the abbreviation "TLS" is used without definition.

(13) In page 1, right column, line 2, "NV" should read "nitrogen-vacancy (NV)", as the abbreviation "NV" is used for first time in the main text. On the other hand, within the abstract, "NV" is defined but never used, so can be left out.

(14) In Fig. 2(b), " $1.43 \pm 0.17 \mu\text{s}$ " in the figure should read " $1.43 \pm 0.17 \text{ ms}$ ".

(15) In Ref. 36, "arXiv:1611.01515" has been published in "Journal of Optics 19, 044003 (2017)".

(16) In Ref. 40, the publication year (2004) is missing.

- (17) In Ref. 43, the journal page should be "227" or "227–234".
- (18) In Fig. S4, the label for the horizontal axis should read " Ω_2/Ω_1 " (" Ω_1/Ω_2 " is larger than 1).
- (19) In page 10 of the supplementary, "Furthermore, it seems that Ω_2 introduces more noise to the system as g does." should read "Furthermore, it seems that Ω_2 introduces more noise to the system than g does."

Reply to referee 3

We are grateful for the overall positive and detailed report from the referee. We highly value his/her constructive comments to improve the readability of our manuscript by suggesting detailed changes in the notation and we also would like to thank the referee for reviewing the subtle details clarifying and improving the scientific content. In what follows, we elaborate a detailed reply on each comment and mark the changes made to the manuscript.

Comment 3.1: *"In the main text, there are seven "see supplementary" instructions without mentioning which part of the relatively long 14-page supplementary to look at. This is very unkind and even irritating, especially when the contents of the supplementary are not ordered in accordance with the main text. For instance, in the second "see supplementary" instruction about analog pulsed version of the scheme (page 2), one has to look at the very last section of the supplementary (Sec. IX, page 13). The instructions should say "see supplementary Sec. xx". In addition, the equations and figures in the supplementary are confusingly numbered 1, 2, 3... as in the main text. They should be labeled as S1, S2, S3... for instance. In the following, I will use these labels to avoid confusion."*

Response: We thank the referee for these valuable suggestions to improve the readability of the manuscript.

Changes made: We followed the suggestions and added to each reference to the supplementary part a more specific instruction, pointing to the relevant section or figure in the supplementary. Moreover, in the supplementary we have changed the labelling of the equations and figures, now containing a prefixed 'S'.

Comment 3.2: *"The distinction between the two terms "coherence time" and "lifetime" is unclear. In page 1 of the main text, "lifetime" means T_1 , while in the supplementary the quantities that were called "coherence times" in the main text are called "lifetimes". It may be that the distinction is subtle in dealing with the dressed states, but still, the interchangeable use of them should be avoided. Moreover, the term "the lifetime of the signal" appearing both in the main text and in the supplementary is incomprehensible to me. The coherence time (or the lifetime as the authors say) measured under non-zero g is still that of the sensor with the signal effectively working as the third drive (say, the triply-dressed state). On the other hand, the signal here is externally applied and sustained during the measurement. Why do the authors use the term "the lifetime of the signal"?"*

Response: We agree that the main text and the supplementary treated the terms 'lifetime' and 'coherence time' in a confusing and to some extent wrong way. In combination with the modifications upon Comment 3.3, we now refer only to the 'coherence time' and indicate every time under which drive (and signal) configuration the respective value was obtained.

Changes made: see Comment 3.3.

Comment 3.3: *“There are four important timescales extracted in the present work: (i) the coherence time under single drive with $g = 0$, (ii) the coherence time under single drive with nonzero g , (iii) the coherence time under double drive with $g = 0$, (iv) the coherence time under double drive with nonzero g . In addition to the mixed use of “coherence time” and “lifetime” as mentioned above, the notations to these timescales are not consistent. The authors interchangeably use “ T_{coh} ” (Fig.s 2, 3), “ $T_{\Omega_1/\Omega_1,\Omega_2}$ ” (page 3 of the main text), “ T_2 ” (Sec. VIII of the supplementary).*

I suggest that they be given separate and systematic notations to avoid confusions. It would be unmistakable if, for instance, they are explicitly defined as T_{Ω_1} for (i), $T_{\Omega_1,g}$ for (ii), T_{Ω_1,Ω_2} for (iii), and $T_{\Omega_1,\Omega_2,g}$ for (iv), and denoted so consistently throughout the main text and the supplementary. In the following, I will use these notations to avoid confusion.”

Response: We agree that a more precise notation under which conditions the coherence time was measured is beneficial in terms of clarity. To mark that the measured coherence times relate to the transverse relaxation time of the sensor, T_2 , we keep the subscript and incorporate in the superscript the suggestion of the referee. Now the four mentioned timescales above are denoted by (i) $T_2^{\Omega_1}$, (ii) $T_2^{\Omega_1,g}$, (iii) $T_2^{\Omega_1,\Omega_2}$ and (iv) $T_2^{\Omega_1,\Omega_2,g}$.

Changes made: We explain in the supplementary Sec. I, second paragraph after equation (S3), the notation for a coherence time measurement under the respective conditions:

“Note, that throughout this manuscript and in the main text, we use the following notation: T_1 , known as the longitudinal relaxation time of the qubit, is called the lifetime of the sensor. T_2^ describes the pure dephasing time of the sensor, if no protection or drive is applied to the sensor. T_2 denotes the transverse relaxation time, which is the coherence time in a pulsed dynamical decoupling experiments. $T_2^{\Omega_1}$ expresses the coherence time under drive Ω_1 and $T_2^{\Omega_1,g}$ characterize a coherence time under drive Ω_1 with an externally applied signal g .”*

The fourth sentence in the supplementary Sec. VI.A was adapted to:

“Fig. S3 shows Rabi (= single drive) measurements with different drive strength Ω_1 displayed against the extracted coherence time of the sensor under drive, $T_2^{\Omega_1}$. It becomes obvious that for a slower drive (< 1 MHz) the coherence time of the sensor, $T_2^{\Omega_1}$, is not increasing monotonically.”

Fig. S3 in the supplementary was adjusted in the y-label, which is called now

“Coherence time under drive $T_2^{\Omega_1}$ (μs)”

and the caption of Fig. S3 was changed to:

“Measure the coherence time of the qubit under an increasing drive, Ω_1 , (essentially, just a Rabi measurement by increasing the Rabi frequency.)”

The y-label in Fig. S4 was adjusted to

“Coherence time under drive $T_2^{\Omega_1, \Omega_2}$ (μs)”

and the caption for this Fig. S4 reads now

“Vary the second drive, Ω_2 , and record the coherence time, $T_2^{\Omega_1, \Omega_2}$, of the qubit, where essentially a double drive measurement without an external signal was performed. The optimal second drive, Ω_2 , maximizes the coherence time $T_2^{\Omega_1, \Omega_2}$ of the sensor.”

The name for Sec. VII in the supplementary part was altered to

“VII. Determine the coherence time of the single drive under an increasing signal”

and the caption text for Fig. S5 was changed to

“Display of the qubit’s coherence time at a fixed drive, Ω_1 , and with an increasing signal strength g .”

On Page 4 in the main text, left column, the third paragraph reads now:

“To examine the signal protection effect in more detail, the coherence time of the sensor is measured as a function of signal strength, g , in a single drive configuration (see also supplementary Sec. VII).”

Comment 3.4: *“From the manuscript, it is difficult to judge what is conceptually new in the present work. In particular, the statement in page 1, “we propose and experimentally demonstrate a sensing scheme that is capable of probing high frequency signals with unprecedented sensitivity”, is vague. The central idea of the present work, concatenated continuous dynamical decoupling, has already been discussed in detail in Ref. 36. It seems to me that the only ingredient added in the present work is the observation that the signal phase can be unknown. Indeed, apart from the sensitivity, the single drive is sufficient to demonstrate the GHz sensing (Fig. 3). In other words, if one regards the second drive in Ref. 36 as a signal, the GHz sensing has already been achieved (in my understanding, the second drive used in Ref. 36 is not phase-matched to the first drive, so it can be regarded as a signal). The manuscript should be revised to clearly separate what is really new in the present work from what has already been known or demonstrated.”*

Response: We would like to refer to the answers of Comments 2.1 and 2.7.

Changes made: We would like to refer to the changes made upon Comments 2.1 and 2.7.

Comment 3.5: *“The authors claim ”that by using a concatenation of two drives, an improvement in coherence time of the sensor by more than one order of magnitude is achieved” (page 2, right). By this, I presume that the authors refer to the results that $T_{\Omega_1} = 40 \mu\text{s}$ for $\Omega_1 = 3.363 \text{ MHz}$ (Fig. S3) and $T_{\Omega_1, \Omega_2} = 400 \mu\text{s}$ for $\Omega_1 = 3.363 \text{ MHz}$ and $\Omega_2 = 505 \text{ kHz}$ (Fig. S4). It may be that the authors also include $T_{\Omega_1, \Omega_2, g} = 1.43 \text{ ms}$ for $\Omega_1 = 3.363 \text{ MHz}$, $\Omega_2 = 505 \text{ kHz}$, and $g = g/4 = 17.3 \text{ kHz}$ as the improvement. However, this is not easy to recognize, because in the main text the results for different conditions are discussed ($T_{\Omega_1} = 60 \mu\text{s}$ for $\Omega_1 = 3.002 \text{ MHz}$ and $T_{\Omega_1, \Omega_2} = 400 \mu\text{s}$).”*

Response: We think that it is a valid point to mention the improvement in coherence time of $T_2^{\Omega_1, \Omega_2, g} = 1.43 \text{ ms}$. We choose to make a rather general statement in the introduction part of the manuscript linking the total improvement in coherence time under a signal g to the improvement in bandwidth for high frequency sensing in general, thus, avoiding the necessity for introducing the notation for coherence times in the introduction part. Given the title of the manuscript, we believe also that it is more beneficial to mention the total magnitude of improvement (which follows from T_2^* to $T_2^{\Omega_1, \Omega_2, g}$) in bandwidth rather than in coherence time. Later in the 'Results' section, we are mentioning and discussing the exact numbers.

In addition, we specify the coherence time under two drive fields more precisely to be $T_{\Omega_1, \Omega_2} \approx 393 \mu\text{s}$ (and not as before as "about 400 μs ") and we refer to this value in the main text and in the supplementary.

Changes made: At first, we add in the last paragraph of the introduction section in the main text (on page 2) a more general statement to describe the total achievement in this work and relate it also to the title:

“Taking into account the effect of an external signal, g , on the sensor during a concatenation of two drive fields, we obtain an improvement in bandwidth for high frequency sensing by three orders of magnitude in comparison to the relaxometry approach.”

A more detailed reflection on the improvement is given in the 'Results' section of the main text, page 3 right column:

“These illustrate a significant increase of the coherence time of the sensor by two orders of magnitude, from $T_2^{\Omega_1} \approx 60 \mu\text{s}$ to a lifetime limited coherence time of ($T_1/2 \approx$) $T_2^{\Omega_1, \Omega_2, g} \approx 1.43 \text{ ms}$. It should be noted that the signal itself can be considered as an additional drive (cf. equation (3)), correcting external errors $\delta\Omega$ of the previous drive and thereby prolonging the coherence time even further. Consequently, we can improve the bandwidth for high frequency sensing by almost three orders of magnitude from $\sim 900 \text{ kHz}$ (for $T_2^ \approx 1.1 \mu\text{s}$) to $\sim 700 \text{ Hz}$ (for a $T_2^{\Omega_1, \Omega_2, g} \approx 1.43 \text{ ms}$). Moreover, in the supplementary Sec. III we discuss an improved version of our scheme which has the potential to*

push the coherence time of the sensor further towards the lifetime limit.”

Comment 3.6: *“Moreover, in Ref. 36, the extension of the coherence time from 2.3 μ s (single drive) to 21 μ s (double drive) has been demonstrated. While the present work has achieved longer timescale than did in Ref. 36, the degree of the extension is almost the same.”*

Response: Comparing the degree of extension in coherence time, the present work indeed reaches similar performance compared to Ref. 36. However, the scheme in Ref. 36 was not investigated in the context of sensing and the drive was incomplete and led to an incoherent drive of all appearing energy levels. (see also the answer to Comment 2.1). Furthermore, in Ref. 36, the T_1 limit of the sensor was not reached, whereas the present scheme does. Even so, we believe that our significant achievement is the first demonstration that dynamical decoupling can be integrated in the task of sensing high frequency signals.

Changes made: Comment 2.1 states the changes to stress the difference between Ref. 36 and our manuscript. In addition, the lifetime limited improvement was already mentioned in the previous Comment 3.5.

Comment 3.7: *“The authors should explain what is really new about the improved coherence time in the preset work, and present experimental data (preferably in the main text) to fully justify the claim.”*

Response: see Comment 2.7.

Changes made: see Comment 2.7.

Comment 3.8: *“Even though the title claims ”narrow-bandwidth”, the authors have made no mention of the bandwidth of the protocol in the main text or the supplementary. The discussion on the attainable and achieved bandwidths, based on both theory and experiments, must be provided.”*

Response: In the conclusion of the main text, we mention the resolution of this scheme, which is the bandwidth. The word bandwidth has not been used, therefore it might have been unclear and led to confusions. We now explain in the introduction (in context of the relaxometry comparison) that the bandwidth is directly linked to the coherence time of the sensor (which we are trying to prolong). In addition, we add to the 'Results' section a specific statement about the improvement in bandwidth.

Changes made: On page 1 of the main text, left column, forth paragraph, we include after the first sentence

“Unlike relaxation measurements comprising a bandwidth $\propto 1/T_2^*$, determined by the pure dephasing time, T_2^* , of the sensor (up to the MHz range), our protocol overcomes the imposed limitation by protecting the addressed two-level system (TLS) with an adapted concatenated CDD approach.”

We add to the 'Results' section of the main text the last sentence before the second paragraph in the right column:

“Consequently, we can improve the bandwidth for high frequency sensing by almost three orders of magnitude from ~ 900 kHz (for $T_2^* \approx 1.1 \mu\text{s}$) to ~ 700 Hz (for a $T_2^{\Omega_1, \Omega_2, g} \approx 1.43$ ms).”

Comment 3.9: “In Fig. 3, $T_{\text{coh}} = 60 \mu\text{s}$ for single drive and $T_{\text{coh}} = 1.43$ ms for double drive are stated. Based on the numbers given, I presume that they mean $T_{\Omega_1} = 60 \mu\text{s}$ and $T_{\Omega_1, \Omega_2, g} = 1.43$ ms, respectively. On the other hand, according to Sec. V of the supplementary, Fig. 3 is obtained by the slope measurements as varying g . Then, it seems appropriate to compare $T_{\Omega_1, g} = 193 \mu\text{s}$ and $T_{\Omega_1, \Omega_2, g} = 1.43$ ms (even more confusingly, the caption says “ $T_{\text{coh}} = 60 \mu\text{s}$ in the signal drive case”). Why do the authors try to compare between $T_{\Omega_1} = 60 \mu\text{s}$ and $T_{\Omega_1, \Omega_2, g} = 1.43$ ms in Fig. 3?”

Response: We believe that the confusions arise in first place due to an ambiguous notation of T_{coh} (and hence Comment 3.3 is partially connected). In principle, we compare $T_2^{\Omega_1, g} = 193 \mu\text{s}$ and $T_2^{\Omega_1, \Omega_2, g} = 1.43$ ms as the referee correctly suggested, i.e. we apply in the single drive case a signal of $g = 26.9$ kHz = $2 \cdot g'$ and in the double drive case a signal of $g = 69.2$ kHz = $4 \cdot g''$. But we do not perform the single drive measurement at the prolonged (optimal) coherence time of the sensor (i.e. at $\tau = T_2^{\Omega_1, g}$ but at $\tau = T_2^{\Omega_1}$) and therefore we neglect naively in the single drive the impact of the signal, g , on the sensor (like it was commonly done in other works). Thereby, we want to show explicitly, that neglecting the coherence time prolonging effect of the signal will result in an even poorer sensitivity. In Fig. 4 we show that we have to include the non-linear impact of the signal onto the sensitivity measurement.

We realize that this was not stated clear enough in our manuscript, which led us to change considerably the caption of Fig. 3 and the explanation in the 'Results' section. Please note that Comment 2.4 is also related.

Changes made: The changes made with respect to this comment are denoted in Comment 2.4.

Comment 3.10: “In Fig. S5, the dashed horizontal line is drawn at $T_{\Omega_1, g}$ (label as “lifetime of signal g ”) $\approx 50 \mu\text{s}$ and denoted as “lifetime without g ”. Where does this value come from? In Fig. S4, T_{Ω_1} (labeled as “lifetime of Ω_1 ”) is $\approx 60 \mu\text{s}$ at $\Omega_1 = 3.002$ MHz, which is consistent with the main text and should appear in Fig. S5. As the dashed horizontal line in Fig. S5 is translated into that in Fig. 4 (indicated as “sensitivity with $g = 0$ ”), the reason for using $50 \mu\text{s}$ rather than the naively expected $60 \mu\text{s}$ should be explained.”

Response: We like to thank the referee for his/her remark about the wrong appearance of the dashed horizontal pink line in Fig. S5 and consequently in Fig. 4.

The measurement errors in the first three points were in fact larger due to a larger fit error. This is because the decaying sinus fit contains less than one period in the measurement leading to an increased error bar in the fit. On

top of that, we performed the measurement in an undersampled way, so that solely the signal g' becomes visible in the data. If during the measurement of Fig. S5 the energy separation ω_0 between the bare states changes (e.g. due to temperature fluctuations of the setup), we would obtain a different Ω_1 and the undersampling reveals a beating in the measurement reflecting the slight deviation from Ω_1 . This beating may become problematic for very low signal strength, g , since it distorts the measurement of g' and both oscillations (the beating and the g') perform less than one full 2π oscillation, since the coherence time decreases.

Remeasuring Ω_1 for every measurement point would result in fact in a larger variation of Ω_1 unless the oscillations are measured with enough points to minimize the fitting error but will considerably increase the overall measurement time. In our measurements, it turned out that it is more favorable to choose a certain Ω_1 and perform all measurement points for that value and consider instead a larger error in the first points.

Changes made: In the main text we modified the caption of Fig. 4 to:

“Projected sensitivity of the single drive scheme (from Fig. S5 in the supplementary) as a function of signal strength g , where the magenta dashed line indicates the sensitivity of the sensor if no signal is applied. The figure illustrates that an external signal has a non-linear effect on the sensitivity of the sensor, which has to be taken into account in the sensitivity estimation.”

In the supplementary, Sec. VII., we add from the second sentence on the following detailed explanation:

“For each measurement point a decaying trace was recorded, which was (under)sampled at multiples of $\tau_{\Omega_1} = 2\pi/\Omega_1$, i.e. we measure at times $t = N\tau_{\Omega_1}$ ($N \in \mathbb{N}$), which removes the effect of Ω_1 on the recorded traces. Consequently, g' was obtained by fitting the data to an exponential decaying sine function. For small values of g , recording the oscillations of g' , and subsequently the coherence time proved to be a challenging measurement, mainly because the period of g' was longer than the coherence time $T_2^{\Omega_1, g}$, so it was difficult to distinguish the two. This caused the measured coherence time to be shorter than expected, given we would expect the coherence time at small g to converge with the measurements without signal (Fig. S3), thus having larger error bars to compensate the ambiguity.”

In Fig. S5 in supplementary and Fig. 4 in the main text the dashed pink coherence time was corrected to display now the value of $\sim 60 \mu\text{s}$.

Comment 3.11: *“In the main text, the term “double dressed state” is used, while “double dressed state”, “double-dressed state”, and “doubly-dressed state” appear randomly in the supplementary. Unify the expression. I feel “doubly-dressed state” most accurate grammatically.”*

Response: We agree that the expressions have to be unified and that "doubly-dressed state" is the proper choice.

Changes made: All terms in the supplementary and the main text are unified to the notation "doubly-dressed state".

Comment 3.12: *"In page 1, left column, line 2, "of a great significance" should read "of great significance"."*

Response: That is correct, we change that in the first sentence.

Changes made: Now the first sentence, page 1, left column reads

"Improving the sensitivity of high frequency sensing schemes is of great significance, especially for classical fields sensing [1-3], detection of electron spins in solids [4,5] and nuclear magnetic resonance spectroscopy [6]."

Comment 3.13: *"In page 1, left column, line 4, the defined abbreviation "NMR" has never been used in the rest of the manuscript. It can be left out."*

Response: We drop the abbreviation "NMR".

Changes made: Compare Comment 3.12 which includes the changes.

Comment 3.14: *"In page 1, left column, line 31, the abbreviation "TLS" is used without definition."*

Response: See Comment 2.2.

Changes made: See Comment 2.2.

Comment 3.15: *"In page 1, right column, line 2, "NV" should read "nitrogen-vacancy (NV)", as the abbreviation "NV" is used for first time in the main text. On the other hand, within the abstract, "NV" is defined but never used, so can be left out."*

Changes made: We leave out the abbreviation "NV" in the abstract and use instead the suggested term "nitrogen-vacancy". We introduce the abbreviation in the first sentence of the fifth paragraph on page 1:

"We demonstrate the performance of continuous dynamical decoupling by applying it to a nitrogen-vacancy (NV) center in diamond with natural abundance of ^{13}C (cf. Fig. 1a)."

Comment 3.16: *"In Fig. 2(b), "1.43 ± 0.17 μs" in the figure should read "1.43 ± 0.17 ms."*

Response: We would like to refer to Comment 1.2.

Changes made: We would like to refer to Comment 1.2.

Comment 3.17: *“In Ref. 36, ”arXiv:1611.01515” has been published in ”Journal of Optics 19, 044003 (2017).”*

Response: Thanks for mentioning. We have replaced the preprint version by the published one.

Changes made: In the revised manuscript Ref. 38 read now:

“J. Teissier, A. Barfuss, and P. Maletinsky, Journal of Optics 19, 044003 (2017).”

Comment 3.18: *“In Ref. 40, the publication year (2004) is missing.”*

Changes made: The missing year was added to Ref. 43 (which was formally Ref. 40). Now we have

“F. Jelezko, T. Gaebel, I. Popa, A. Gruber, and J. Wrachtrup, Physical Review Letters 92, 076401 (2004).”

Comment 3.19: *“In Ref. 43, the journal page should be ”227” or ”227–34”.”*

Response: In fact, our bibtex file contains the journal page range, but the revtex4-1 class in L^AT_EX (specifically the bibtex style) lacks the ability for displaying it correctly according to the criteria of Nature Communications. Therefore we changed the document class of the L^AT_EX-file to the nature style, which provides the correct display of citations. This version will be submitted to the editor. However, the version for the referees will be written in the revtex4-1 class and cannot contain the requested changes.

Comment 3.20: *“In Fig. S4, the label for the horizontal axis should read ” Ω_2/Ω_1 ” (” Ω_1/Ω_2 ” is larger than 1).”*

Response: That is true, the horizontal axis label contained a mistake.

Changes made: We correct the x-label of Fig. S4 to

“Ratio Ω_2/Ω_1 (with $\Omega_1/2\pi = 3.363$ MHz)”

Comment 3.21: *“In page 10 of the supplementary, ”Furthermore, it seems that Ω_2 introduces more noise to the system as g does.” should read ”Furthermore, it seems that Ω_2 introduces more noise to the system than g does.””*

Changes made: We corrected the sentence as suggested to:

“Furthermore, it seems that Ω_2 introduces more noise to the system than g does.”

List of changes

Corrections not mentioned by the referees but realized in the review process by the authors:

- In supplementary in Sec. I., II. and III., we specify explicitly the relative error terms of the respective drive with the appropriate subscript, δ_{Ω_1} and δ_{Ω_2} .
- Fig. 1a in Supplementary was not saved correctly, i.e. just the fit of the ODMR measurement could be seen and the data points were disappeared. We now saved the figure correctly.
- A closing bracket ')' is missing at the end of the caption in Fig. S1 in Supplementary.
- The bibliography is now present in a separate bibliography file to have a more flexible adjustment of the final bibliography.
- Add in supplementary before equation (S10) to the equation label a parenthesis
“...from equation S9...” \Rightarrow “...from equation (S9)...”.
- Acknowledge the software suite Qudi in the main text.
- We add the Ref.s 19, 20, 39, 45 and 47 to our manuscript.

Narrow-bandwidth sensing of high-frequency fields with continuous dynamical decoupling

A. Stark,^{1,2} N. Aharon,³ T. Uden,² D. Louzon,^{2,3} A. Huck,¹ A. Retzker,³ U.L. Andersen,¹ and F. Jelezko^{2,4}

¹Department of Physics, Technical University of Denmark, Fysikvej, Kongens Lyngby 2800, Denmark

²Institute for Quantum Optics, Ulm University, Albert-Einstein-Allee 11, Ulm 89081, Germany

³Racah Institute of Physics, The Hebrew University of Jerusalem, Jerusalem 91904, Israel

⁴Center for Integrated Quantum Science and Technology (IQST), Ulm University, 89081 Germany

State-of-the-art methods for sensing weak AC fields are only efficient in the low frequency domain (< 10 MHz). The inefficiency of sensing high frequency signals is due to the lack of ability to use dynamical decoupling. In this paper we show that dynamical decoupling can be incorporated into high frequency sensing schemes and by this we demonstrate that the high sensitivity achieved for low frequency can be extended to the whole spectrum. While our scheme is general and suitable to a variety of atomic and solid-state systems, we experimentally demonstrate it with the nitrogen-vacancy center in diamond. For a diamond with natural abundance of ^{13}C we achieve coherence times up to 1.43 ms resulting in a smallest detectable magnetic field strength of 4 nT at 1.6 GHz. Attributed to the inherent nature of our scheme, we observe an additional increase in coherence time due to the signal itself.

PACS numbers: 76.30.Mi, 76.90.+d, 07.55.Ge, 03.65.Yz, 03.67.Pp

Keywords: dynamical decoupling, high frequency, quantum sensing, two-level system, nitrogen-vacancy, diamond

Improving the sensitivity of high frequency sensing schemes is of great significance, especially for classical fields sensing [1–3], detection of electron spins in solids [4, 5] and nuclear magnetic resonance spectroscopy [6]. The common method to detect high frequency field components is based on relaxation measurements, where the signal induces an observable effect on the lifetime, T_1 , of the probe system [4, 5, 7]. Nevertheless, the sensitivity of this method is limited by the pure dephasing time T_2^* of the probe system.

Pulsed dynamical decoupling [8–10] can substantially increase the coherence time [11–18]. In order to carry out sensing with a decoupling scheme, the frequency of the decoupling pulses has to be matched with the frequency of the target field [19, 20]. This largely restricts the approach to low frequencies, as the repetitive application of pulses is limited by the maximum available power per pulse [21]. The same power restrictions are present for very rapid and composite pulse sequences aimed to decrease both external and controller noise [22–26].

With continuous dynamical decoupling (CDD) [21, 27–38] robustness to external and controller noise can be attained, especially for multi-level systems [39–42]. However, the significance of CDD for sensing high frequency fields remained elusive. Indeed, it was unclear whether it is possible to incorporate such a protection into the metrology task of sensing frequencies in the GHz domain. The first step towards this goal was done recently by integrating CDD in the sensing of high frequency fields with three level systems [42].

In this article, we propose, analyze and experimentally demonstrate for the first time a sensing scheme that is capable of probing high frequency signals with a coherence time, T_2 , limited sensitivity. Unlike relaxation measurements comprising a bandwidth $\propto 1/T_2^*$, determined by the pure dephasing time, T_2^* , of the sensor (up to the MHz range), our protocol overcomes the imposed limitation by protecting the addressed two-level system (TLS) with an adapted concatenated CDD

Figure 1. Schematic representation of our setup: (a) The NV center probes an external signal while it is being manipulated by the control fields. (b) Schematic representation of the sequence applied in this work. (c) The protected TLS: The bare system, H_0 , is subjected to strong environmental noise δB . Applying a strong drive, Ω_1 , opens a protected gap, now subjected mainly to drive fluctuations $\delta\Omega_1$. A second drive, Ω_2 , is then applied to protect the TLS, H_I , from these fluctuations, resulting in a TLS, H_{II} , on resonance with the signal, $g'' = g/4$, with noise mainly from the second weak drive $\delta\Omega_2 \ll \delta\Omega_1$.

approach. We use and adjust it such that high frequency sensing becomes feasible even for not phase-matched signals. As a result, the proposed scheme is generic and works for many atomic or solid state TLS, in which the energy gap matches the frequency of the signal under interrogation. A remarkable feature of our scheme is the fact that the signal to be probed also works partially as a decoupling drive and thus improves further the sensitivity of the sensor.

We demonstrate the performance of continuous dynamical decoupling by applying it to a nitrogen-vacancy (NV) center in diamond with natural abundance of ^{13}C (cf. Fig. 1a). Here, we utilize two of its ground sub-levels as the TLS. The states of the NV center can be read out and initialized by a 532 nm

laser, which reveals spin dependent fluorescence between the two levels [43–45]. The system can be manipulated by driving it with microwave fields. We show that by using a concatenation of two drives, an improvement in coherence time of the sensor by more than one order of magnitude is achieved. Taking into account the effect of an external signal, g , on the sensor during a concatenation of two drive fields, we obtain an improvement in bandwidth for high frequency sensing by three orders of magnitude in comparison to the relaxometry approach. Moreover, we report on the measurement of a weak high frequency signal with strength g , which relates to a smallest detectable magnetic field amplitude of $\delta B_{\min} \approx 4$ nT.

THE SENSING SCHEME

The basic idea of utilizing concatenated continuous driving to create a robust qubit is illustrated in Fig. 1b. The concatenation of two phase-matched driving fields results in a robust qubit [36, 42]. In what follows we show that such a robust qubit can be utilized as a sensor for frequencies in the range of the qubit's energy separation and hence, dynamical decoupling can be integrated into the sensing task.

By the concatenated driving, the qubit is prepared in a state that allows for strong coherent coupling to the high-frequency signal to be probed (corresponding to the last TLS in Fig. 1c). In the total Hamiltonian, H , we consider the concatenation of two driving fields of strength (the Rabi frequency) Ω_1 and Ω_2 , respectively. The Hamiltonians of the TLS, H_0 , the protecting driving fields, $H_{\Omega_1}, H_{\Omega_2}$ and the signal, H_s , are given by

$$H = H_0 + H_{\Omega_1} + H_{\Omega_2} + H_s = \frac{\omega_0}{2} \sigma_z + \Omega_1 \sigma_x \cos(\omega_0 t) + \Omega_2 \sigma_y \cos(\omega_0 t) \cos(\Omega_2 t) + g \sigma_x \cos(\omega_s t + \varphi), \quad (1)$$

where ω_0 is the energy gap of the bare states ($\hbar = 1$), ω_s is the frequency of the signal, and g is the signal strength which we want to determine. We tune the system, i.e., ω_0 , Ω_1 , and Ω_2 , such that $\omega_s = \omega_0 + \Omega_1 + \Omega_2/2$.

It is an important feature that phase matching between the signal and the control is not required, which means that the signal phase φ can be unknown and moreover, it may vary between experimental runs. In addition, we make the assumption that $\omega_0 \gg \Omega_1 \gg \Omega_2 \gg g$. Moving to the interaction picture (IP) with respect to $H_0 = \frac{\omega_0}{2} \sigma_z$ and making the rotating-wave-approximation, we obtain

$$H_I = \frac{\Omega_1}{2} \sigma_x + \frac{\Omega_2}{2} \sigma_y \cos(\Omega_1 t) + \frac{g}{2} \left(\sigma_+ e^{-i((\Omega_1 + \Omega_2/2)t + \varphi)} + \sigma_- e^{+i((\Omega_1 + \Omega_2/2)t + \varphi)} \right). \quad (2)$$

This picture incorporates the effect of Ω_1 onto a TLS and express the new system in eigenstates of σ_x , the $|\pm\rangle$ (dressed) states, which separates the contributions from Ω_2 and g . For a large enough drive Ω_1 , the $|\pm\rangle$ eigenstates are decoupled (in first order) from magnetic noise, $\delta B \sigma_z$, because $\langle \pm | \sigma_z | \pm \rangle = 0$. However, power fluctuations $\delta \Omega_1$ of Ω_1 limit the coherence

time of the dressed states. The resulting IP is illustrated in the second TLS in Fig. 1c.

We continue by moving to a second IP with respect to $H_{01} = \frac{\Omega_1}{2} \sigma_x$, which leads to

$$H_{II} = \frac{\Omega_2}{4} \sigma_y + \frac{g}{4} \left(-i \sigma_+ e^{-i(\frac{\Omega_2}{2}t + \varphi)} + i \sigma_- e^{+i(\frac{\Omega_2}{2}t + \varphi)} \right). \quad (3)$$

Once again, we incorporate Ω_2 into the dressed states, so that solely the contribution of the signal g becomes obvious, which is depicted in the last TLS of Fig. 1c. The second drive, Ω_2 , which is larger than $\delta \Omega_1$, creates effectively doubly-dressed states (the σ_y eigenstates). These doubly-dressed states are immune to power fluctuations of Ω_1 and hence prolong the coherence time (see Supplementary Sec. II for more details). Moving to the third IP with respect to $H_{02} = \frac{\Omega_2}{4} \sigma_y$ results in

$$H_{III} = \frac{g}{8} (\sigma_+ e^{-i\varphi} + \sigma_- e^{+i\varphi}), \quad (4)$$

where we can clearly see that the signal g induces rotations in the robust qubit subspace (either with σ_+ or σ_-). These rotations are obtained for any value of an arbitrary phase φ and the bandwidth ($\ll 1/T_2$) is now limited by the coherence time, T_2 , of the sensor. Hence, if a given TLS exhibits the possibility of manipulating it via drive fields H_{Ω_1} and H_{Ω_2} , we can achieve a high frequency sensor in the range of ω_0 .

By this, we overcome the low frequency limit that is common to state-of-the-art pulsed dynamical decoupling sensing methods. In addition, we present an analog pulsed version of our scheme, where the pulsing rate is much lower than the frequency of the signal (see Supplementary Sec. IX). However it is not a direct measurement of the signal, but based on a signal demodulation approach. Compared to the pulsed schemes, continuous dynamical decoupling does not suffer from being susceptible to higher harmonics of the decoupling window appearing naturally from the periodic character of the pulsed sequence [46]. Eventually, less power per unit time is used in the continuous scheme leading to a smaller overall noise contribution from the drive.

RESULTS

After determining the optimal drive parameters, Ω_1 and Ω_2 , for the concatenated sensing sequence, and thereby maximize the coherence times, $T_2^{\Omega_1}$ and $T_2^{\Omega_1, \Omega_2}$, respectively, of the sensor (see Supplementary Sec. VI), we apply an external high frequency signal (according to H_s in equation (1)) tuned to one of the four appearing energy gaps ω_s of the doubly-dressed states. In these energy gaps an effective population transfer can occur between the states of the robust TLS, evidenced by signal induced Rabi oscillations at a rate $g'' = g'/2 = g/4$ in the double drive case (see Supplementary Sec. II).

The measurements take place in the laboratory frame, i.e. all three contributions Ω_1, Ω_2 and g to the population dynamics of the TLS will be visible. In order to see solely the

Figure 2. Measurements of an external signal of strength g . (a) In a single drive approach with $\Omega_1/2\pi = 3.002$ MHz a signal $g' = g/2$ is recorded. (b) By the application of two drive fields with $\Omega_1/2\pi = 3.363$ MHz and $\Omega_2/2\pi = 505$ kHz, we record a signal $g'' = g/4$ and increase the coherence time of the sensor by one order of magnitude with respect to the case of $g = 0$.

effect of g on the TLS, we alter the modulation of the second drive in equation (1) to $\cos(\Omega_1 t + \pi/2)$. This does not change the performance of the scheme, but only changes the axis of rotation to σ_z for the second drive Ω_2 . Since the readout laser is effectively projecting the population in the σ_z eigenbasis, we can make the Rabi rotations of Ω_2 invisible to the readout. To remove the effect of Ω_1 in the data, we can simply sample the measurement at multiple times of $\tau_{\Omega_1} = 2\pi/\Omega_1$, i.e. we measure at times $t = N\tau_{\Omega_1}$ ($N \in \mathbb{N}$). This procedure reveals directly $g''(g')$ as the signal induces Rabi oscillations of the robust qubit under double (single) drive (Fig. 2). Alternatively, we could have applied at the end of the drive a correction pulse in order to complete the full Ω_1 and Ω_2 rotations so that just the effect of g'' remains.

Without a signal, g , we achieve coherence times of $T_2^{\Omega_1} \approx$

Figure 3. Comparison of the smallest measurable magnetic field change $\delta B_{\min} = (2\pi\delta g_{\min})/\gamma_{\text{NV}}$ as a function of total measurement time. To show the total improvement, we obtain $\sigma(t)$ at $\tau = T_2^{\Omega_1} \approx 60 \mu\text{s}$ in the single drive case and $\sigma(t)$ at $\tau = T_2^{\Omega_1, \Omega_2, g} \approx 1.43 \text{ ms}$ in the double drive case. Note, that for both data traces a signal was always present, $g/2\pi = 26.9$ kHz and $g/2\pi = 69.2$ kHz in the single and in the double drive, respectively. But only in the double drive the coherence time prolonging effect of g was included into the choice of τ for equation (5) (i.e. the measurement was performed at $\tau = T_2^{\Omega_1, \Omega_2, g}$ instead at $\tau = T_2^{\Omega_1, \Omega_2} \approx 393 \mu\text{s}$).

$60 \mu\text{s}$ with a single drive ($\Omega_2 = 0$, compare Fig. S3 in Sec. VI.A in Supplementary) and $T_2^{\Omega_1, \Omega_2} \approx 393 \mu\text{s}$ with a double drive (compare Fig. S4 in Sec. VI.B in Supplementary). The results for long and slow Rabi oscillations induced by an external signal, g , under single and double drive ($\Omega_2/\Omega_1 \approx 0.15$) are shown in Fig. 2. These illustrate a significant increase of the coherence time of the sensor by two orders of magnitude, from $T_2^{\Omega_1} \approx 60 \mu\text{s}$ to a lifetime limited coherence time of $(T_1/2 \approx) T_2^{\Omega_1, \Omega_2, g} \approx 1.43 \text{ ms}$. It should be noted that the signal itself can be considered as an additional drive (cf. equation 3), correcting external errors $\delta\Omega$ of the previous drive and thereby prolonging the coherence time even further. Consequently, we can improve the bandwidth for high frequency sensing by almost three orders of magnitude from ~ 900 kHz (for $T_2^{\Omega_1} \approx 1.1 \mu\text{s}$) to ~ 700 Hz (for a $T_2^{\Omega_1, \Omega_2, g} \approx 1.43 \text{ ms}$). Moreover, in the Supplementary Sec. III we discuss an improved version of our scheme which has the potential to push the coherence time of the sensor further towards the lifetime limit.

To benchmark the double drive scheme against a standard single drive approach, we determine the smallest magnetic field which can be sensed after an accumulation time t . The smallest measurable signal S is eventually bounded by the smallest measurable magnetic field change δB_{\min} , which is found to be

$$\delta B_{\min}(t, \tau) = \frac{\delta S}{\max \left| \frac{\partial S}{\partial B} \right|} = \frac{1}{\gamma_{\text{NV}}} \frac{\sigma(t)}{\alpha \tau C}. \quad (5)$$

Figure 4. Projected sensitivity of the single drive scheme (from Fig. S5 in the Supplementary) as a function of signal strength g , where the magenta dashed line indicates the sensitivity of the sensor if no signal is applied. The figure illustrates that an external signal has a non-linear effect on the sensitivity of the sensor, which has to be taken into account in the sensitivity estimation.

Here, $\gamma_{\text{NV}}/2\pi = 28.8 \text{ GHz T}^{-1}$ is the gyromagnetic ratio of the NV defect, $\sigma(t)$ is the standard deviation of the measured normalized fluorescence counts after time t , α accounts for a different phase accumulation rate depending on the decoupling scheme and C is the contrast of the signal (see Supplementary Sec. V for detailed derivation). Since the photon counting is shot noise limited, we have $\sigma(t) = 1/\sqrt{N_{\text{ph}} \cdot N}$, with N_{ph} being the number of photons measured in τ and $N = t/\tau$ is the number of sequence repetitions. With this, equation (5) will transform in the commonly known form [47, 48] with some measurement dependent constants.

We recorded $\sigma(t)$ as a function of time and use this to determine δB_{min} . The results of this measurement for both the single and double drive are summarized in Fig. 3. The sensitivity can be obtained by $\eta(\tau) = \delta B_{\text{min}}(t, \tau)\sqrt{t}$, which is optimal in the vicinity of the coherence time of the sensor, $\tau \approx T_2$. With our system, we achieve a sensitivity of $\eta_{\Omega_1, \Omega_2, g} \lesssim 1 \mu\text{T Hz}^{-0.5}$ in the double drive case at $\sim 1.6 \text{ GHz}$, which should be compared to $\eta_{\Omega_1, g} \lesssim 20 \mu\text{T Hz}^{-0.5}$ for a single drive approach.

Both traces in Fig. 3 were recorded while a signal g was applied. Apart from the mere fact, that the number of driving fields are different, the specific choice for τ will also determine the magnitude of the smallest measurable magnetic field change δB_{min} . Obtaining the coherence time without a signal, g , (which are $T_2^{\Omega_1}$ and $T_2^{\Omega_1, \Omega_2}$), is a common practice in the field, but will not result in a correct choice of τ for the sensitivity measurement and also for equation (5), since the signal has an impact on the sensor's sensitivity. However, if δB_{min} shall be evaluated correctly, then the non-linearity, i.e. the coherence time prolonging effect of the signal, has to be taken into account. Otherwise an even worse δB_{min} will be measured as it is exemplarily shown for the single drive case

in Fig. 3, where δB_{min} was evaluated and measured under the naive assumption that the signal has no effect on the coherence time of the sensor (i.e. we measure at $\tau = T_2^{\Omega_1}$ and not at $\tau = T_2^{\Omega_1, g}$). This effect was included in the double drive case.

To examine the signal protection effect more in detail, the coherence time of the sensor is measured as a function of signal strength, g , in a single drive configuration (see also Supplementary Sec. VII). From these measurements we project the sensitivity associated with a specific signal strength (Fig. 4), assuming $\sigma(t)$ is unchanged for the same repetition N . This is a reasonable assumption given that the only difference between measurements is the signal strength, g , and sequence length, τ .

Eventually, this phenomenon, which seems to be an inherent part of this continuous scheme, can be used to further increase the performance of the sensor by fine tuning the controlled parameters (static bias field B_{bias} and thereby changing ω_0 , Ω_1 and Ω_2) to match the signal frequency, ω_s , and strength, g (see Supplementary Sec. III for further discussions).

CONCLUSION

We have demonstrated for the first time that dynamical decoupling can be used in the context of sensing high frequency fields. In contrast to state-of-the-art pulsed dynamical decoupling protocols, we can show that continuous dynamical decoupling can be simultaneously integrated into the sensing task. By utilizing a NV center in diamond we have demonstrated by pure concatenation of two drives a coherence time of $\sim 393 \mu\text{s}$ which constitutes an improvement of more than two orders of magnitude over T_2^* , and an increase of resolution from the MHz to a few kHz. The application of this method for wireless communication [49] could have a transformative effect due to the high resolution of the protocol. Since the protocol is applicable to a variety of solid-state, molecular, and atomic systems, we believe that it has a great potential to have a significant impact on many fields and tasks that involve high frequency sensing (up to frequencies in the THz range). Eventually, this method could also be used to improve the coupling to quantum systems [30].

Acknowledgements

The experiments presented here were supported by the Qudi Software Suite [50]. A. S., A. H. and U. L. A. acknowledge funding from the Innovation Foundation Denmark through the project EXMAD and the Qubiz center, and the Danish Research Council via the Sapere Aude project (DIMS). T. U. and F.J. acknowledge the Volkswagenstiftung. A. R. acknowledges the support of the Israel Science Foundation (grant no. 1500/13), the support of the European commission, EU Project DIADEMS. This project has received

funding from the European Union's Horizon 2020 research and innovation program under grant agreement No 667192 Hyperdiamond and Research Cooperation Program and DIP program (FO 703/2-1).

-
- [1] M. Chipaux, L. Toraille, C. Larat, L. Morvan, S. Pezzagna, J. Meijer, and T. Debuisschert, *Applied Physics Letters* **107**, 233502 (2015).
- [2] S. Kolkowitz, A. Safira, A. A. High, R. C. Devlin, S. Choi, Q. P. Unterreithmeier, D. Patterson, A. S. Zibrov, V. E. Manucharyan, H. Park, and M. D. Lukin, *Science* **347**, 1129 (2015).
- [3] L. Shao, M. Zhang, M. Markham, A. M. Edmonds, and M. Lonar, *Physical Review Applied* **6**, 064008 (2016).
- [4] A. O. Sushkov, N. Chisholm, I. Lovchinsky, M. Kubo, P. K. Lo, S. D. Bennett, D. Hunger, A. Akimov, R. L. Walsworth, H. Park, and M. D. Lukin, *Nano Letters* **14**, 6443 (2014).
- [5] L. T. Hall, P. Kehayias, D. A. Simpson, A. Jarmola, A. Stacey, D. Budker, and L. C. L. Hollenberg, *Nature Communications* **7**, 10211 (2016).
- [6] R. Kimmich and E. Anoardo, *Progress in Nuclear Magnetic Resonance Spectroscopy* **44**, 257 (2004).
- [7] D. Schmid-Lorch, T. Hberle, F. Reinhard, A. Zappe, M. Slota, L. Bogani, A. Finkler, and J. Wrachtrup, *Nano Letters* **15**, 4942 (2015).
- [8] E. L. Hahn, *Physical Review* **80**, 580 (1950).
- [9] H. Y. Carr and E. M. Purcell, *Physical Review* **94**, 630 (1954).
- [10] S. Meiboom and D. Gill, *Review of Scientific Instruments* **29**, 688 (1958).
- [11] L. Viola and S. Lloyd, *Physical Review A* **58**, 2733 (1998).
- [12] M. J. Biercuk, H. Uys, A. P. VanDevender, N. Shiga, W. M. Itano, and J. J. Bollinger, *Nature* **458**, 996 (2009).
- [13] J. Du, X. Rong, N. Zhao, Y. Wang, J. Yang, and R. B. Liu, *Nature* **461**, 1265 (2009).
- [14] G. d. Lange, Z. H. Wang, D. Rist, V. V. Dobrovitski, and R. Hanson, *Science* **330**, 60 (2010).
- [15] C. A. Ryan, J. S. Hodges, and D. G. Cory, *Physical Review Letters* **105**, 200402 (2010).
- [16] B. Naydenov, F. Dolde, L. T. Hall, C. Shin, H. Fedder, L. C. L. Hollenberg, F. Jelezko, and J. Wrachtrup, *Physical Review B* **83**, 081201 (2011).
- [17] Z.-H. Wang, G. de Lange, D. Rist, R. Hanson, and V. V. Dobrovitski, *Physical Review B* **85**, 155204 (2012).
- [18] N. Bar-Gill, L. M. Pham, A. Jarmola, D. Budker, and R. L. Walsworth, *Nature Communications* **4**, 1743 (2013).
- [19] J. M. Taylor, P. Cappellaro, L. Childress, L. Jiang, D. Budker, P. R. Hemmer, A. Yacoby, R. Walsworth, and M. D. Lukin, *Nature Physics* **4**, 810 (2008).
- [20] S. Kotler, N. Akerman, Y. Glickman, A. Keselman, and R. Ozeri, *Nature* **473**, 61 (2011).
- [21] G. Gordon, G. Kurizki, and D. A. Lidar, *Physical Review Letters* **101**, 010403 (2008).
- [22] K. Khodjasteh and D. A. Lidar, *Physical Review Letters* **95**, 180501 (2005).
- [23] G. S. Uhrig, *Physical Review Letters* **98**, 100504 (2007).
- [24] A. M. Souza, G. A. Ivarez, and D. Suter, *Physical Review Letters* **106**, 240501 (2011).
- [25] W. Yang, Z.-Y. Wang, and R.-B. Liu, *Frontiers of Physics in China* **6**, 2 (2011).
- [26] D. Farfurnik, A. Jarmola, L. M. Pham, Z. H. Wang, V. V. Dobrovitski, R. L. Walsworth, D. Budker, and N. Bar-Gill, *Physical Review B* **92**, 060301 (2015).
- [27] F. F. Fanchini, J. E. M. Hornos, and R. d. J. Napolitano, *Physical Review A* **75**, 022329 (2007).
- [28] A. Bermudez, F. Jelezko, M. B. Plenio, and A. Retzker, *Physical Review Letters* **107**, 150503 (2011).
- [29] A. Bermudez, P. O. Schmidt, M. B. Plenio, and A. Retzker, *Physical Review A* **85**, 040302 (2012).
- [30] J. Cai, F. Jelezko, N. Katz, A. Retzker, and M. B. Plenio, *New Journal of Physics* **14**, 093030 (2012).
- [31] X. Xu, Z. Wang, C. Duan, P. Huang, P. Wang, Y. Wang, N. Xu, X. Kong, F. Shi, X. Rong, and J. Du, *Physical Review Letters* **109**, 070502 (2012).
- [32] D. A. Golter, T. K. Baldwin, and H. Wang, *Physical Review Letters* **113**, 237601 (2014).
- [33] P. Rabl, P. Cappellaro, M. V. G. Dutt, L. Jiang, J. R. Maze, and M. D. Lukin, *Physical Review B* **79**, 041302 (2009).
- [34] J. Clausen, G. Bensky, and G. Kurizki, *Physical Review Letters* **104**, 040401 (2010).
- [35] A. Laucht, R. Kalra, S. Simmons, J. P. Dehollain, J. T. Muhonen, F. A. Mohiyaddin, S. Freer, F. E. Hudson, K. M. Itoh, D. N. Jamieson, J. C. McCallum, A. S. Dzurak, and A. Morello, *Nature Nanotechnology* **12**, 61 (2017).
- [36] J.-M. Cai, B. Naydenov, R. Pfeiffer, L. P. McGuinness, K. D. Jahnke, F. Jelezko, M. B. Plenio, and A. Retzker, *New Journal of Physics* **14**, 113023 (2012).
- [37] I. Cohen, S. Weidt, W. K. Hensinger, and A. Retzker, *New Journal of Physics* **17**, 043008 (2015).
- [38] J. Teissier, A. Barfuss, and P. Maletinsky, *Journal of Optics* **19**, 044403 (2017).
- [39] I. Baumgart, J.-M. Cai, A. Retzker, M. Plenio, and C. Wunderlich, *Physical Review Letters* **116**, 240801 (2016).
- [40] N. Timoney, I. Baumgart, M. Johanning, A. F. Varn, M. B. Plenio, A. Retzker, and C. Wunderlich, *Nature* **476**, 185 (2011).
- [41] N. Aharon, M. Drewsen, and A. Retzker, *Physical Review Letters* **111**, 230507 (2013).
- [42] N. Aharon, I. Cohen, F. Jelezko, and A. Retzker, *New Journal of Physics* **18**, 123012 (2016).
- [43] F. Jelezko, T. Gaebel, I. Popa, A. Gruber, and J. Wrachtrup, *Physical Review Letters* **92**, 076401 (2004).
- [44] F. Jelezko, T. Gaebel, I. Popa, M. Domhan, A. Gruber, and J. Wrachtrup, *Physical Review Letters* **93**, 130501 (2004).
- [45] G. Balasubramanian, I. Y. Chan, R. Kolesov, M. Al-Hmoud, J. Tisler, C. Shin, C. Kim, A. Wojcik, P. R. Hemmer, A. Krueger, T. Hanke, A. Leitenstorfer, R. Bratschitsch, F. Jelezko, and J. Wrachtrup, *Nature* **455**, 648 (2008).
- [46] M. Loretz, J. Boss, T. Rosskopf, H. Mamin, D. Rugar, and C. Degen, *Physical Review X* **5**, 021009 (2015).
- [47] W. M. Itano, J. C. Bergquist, J. J. Bollinger, J. M. Gilligan, D. J. Heinzen, F. L. Moore, M. G. Raizen, and D. J. Wineland, *Physical Review A* **47**, 3554 (1993).
- [48] D. Budker and M. Romalis, *Nature Physics* **3**, 227 (2007).
- [49] C. Baylis, M. Fellows, L. Cohen, and R. J. M. II, *IEEE Microwave Magazine* **15**, 94 (2014).
- [50] J. M. Binder, A. Stark, N. Tomek, J. Scheuer, F. Frank, K. D. Jahnke, C. Mller, S. Schmitt, M. H. Metsch, T. Uden, T. Gehring, A. Huck, U. L. Andersen, L. J. Rogers, and F. Jelezko, *SoftwareX* **6**, 85 (2017).

Narrow-bandwidth sensing of high-frequency fields with continuous dynamical decoupling - Supplementary material

A. Stark,^{1,2} N. Aharon,³ T. Uden,² D. Louzon,^{2,3} A. Huck,¹ A. Retzker,³ U.L. Andersen,¹ and F. Jelezko^{2,4}

¹*Department of Physics, Technical University of Denmark, Fysikvej, Kongens Lyngby 2800, Denmark*

²*Institute for Quantum Optics, Ulm University, Albert-Einstein-Allee 11, Ulm 89081, Germany*

³*Racah Institute of Physics, The Hebrew University of Jerusalem, Jerusalem 91904, Israel*

⁴*Center for Integrated Quantum Science and Technology (IQST), Ulm University, 89081 Germany*

I. SENSING OF A HIGH FREQUENCY SIGNAL WITH A SINGLE DRIVE

In this section we describe in detail how the enhanced sensing of high frequency signals can be performed with a two-level system (TLS) by employing a single continuous driving field. The Hamiltonian of the TLS is given by

$$H = \frac{\omega_0}{2} \sigma_z + \Omega_1 \left(1 + \delta_{\Omega_1}(t) \right) \cos(\omega_0 t) \sigma_x + g \cos(\omega_s t + \varphi) \sigma_x + \delta_B(t) \sigma_z, \quad (\text{S1})$$

where ω_0 corresponds to the energy gap of the bare states, Ω_1 is the Rabi frequency of the on-resonance driving field, g is the Rabi frequency of the signal, ω_s is the frequency of the signal, and φ is a random phase, which indicates that the signal field and the driving field are not phased-matched, and therefore φ has a different value in each experiment. We tune the system, i.e., ω_0 and Ω_1 , such that $\omega_s = \omega_0 + \Omega_1$. In addition, $\delta_{\Omega_1}(t)\Omega_1$ represents the fluctuations of the driving field, and $\delta_B(t)$ is the magnetic noise. Moving to the interaction picture (IP) with respect to $H_0 = \frac{\omega_0}{2} \sigma_z$ and, assuming that $\omega_0 \gg \Omega_1$, making the rotating-wave-approximation (RWA) we get

$$H_I = \frac{\Omega_1}{2} \left(1 + \delta_{\Omega_1}(t) \right) \sigma_x + \frac{g}{2} \left(\sigma_+ e^{-i(\Omega_1 t + \varphi)} + \sigma_- e^{+i(\Omega_1 t + \varphi)} \right) + \delta_B(t) \sigma_z. \quad (\text{S2})$$

In the basis of the dressed states, the σ_x eigenstates, the Hamiltonian reads ($x \rightarrow z$, $z \rightarrow -x$, $y \rightarrow y$)

$$H_I = \frac{\Omega_1}{2} \left(1 + \delta_{\Omega_1}(t) \right) \sigma_z + \frac{g}{2} \left(\cos(\Omega_1 t + \varphi) \sigma_z + \sin(\Omega_1 t + \varphi) \sigma_y \right) - \delta_B(t) \sigma_x. \quad (\text{S3})$$

Because the magnetic noise couples between the dressed states, in first order, the noise induces a longitudinal relaxation (decay) rate of $\sim S_{BB}(\Omega_1)$, where S_{BB} is the power spectrum of the noise. A large enough Ω_1 ensures that the longitudinal relaxation rate is negligible ($S_{BB}(\Omega_1) \ll 1/T_1$). In this case, within the experiment time the noise does not induce transitions between the dressed states, but does result in a second order fluctuating phase shift of $\sim \delta_B^2(t)/\Omega_1$. The resulting dephasing rate is considerably diminished with an increasing Ω_1 [1]. The main limitation on the coherence time of the dressed states, $T_2^{\Omega_1}$, is due to power fluctuations of the driving field, $\delta_{\Omega_1}(t)\Omega_1$, which limit the coherence time to $T_2^{\Omega_1} \sim 1/(\delta_{\Omega_1}(t)\Omega_1)$. For typical experimental setups $\delta_{\Omega_1}(t) \sim 0.1 - 1\%$ implies an improvement of ~ 1 order of magnitude in the coherence time compared to T_2^* .

Note, that throughout this manuscript and in the main text, we use the following notation: T_1 , known as the longitudinal relaxation time of the qubit, is called the lifetime of the sensor. T_2^* describes the pure dephasing time of the sensor, if no protection or drive is applied to the sensor. T_2 denotes the transverse relaxation time, which is the coherence time in a pulsed dynamical decoupling experiments. $T_2^{\Omega_1}$ expresses the coherence time under drive Ω_1 and $T_2^{\Omega_1, g}$ characterize a coherence time under drive Ω_1 with an externally applied signal g .

We continue by moving to the interaction picture (IP) with respect to $H_{0I} = \frac{\Omega_1}{2} \sigma_z$ (in the basis of the dressed states) and, assuming that $\Omega_1 \gg g$, making the rotating-wave-approximation (RWA), which leads to

$$H_{II} = \frac{g}{4} \left(-i\sigma_+ e^{-i\varphi} + i\sigma_- e^{+i\varphi} \right) + \frac{\delta_{\Omega_1}(t)\Omega_1}{2} \sigma_z, \quad (\text{S4})$$

where we neglected the (fast rotating) terms of the magnetic noise. The signal in H_{II} corresponds to the on-resonance coupling between the dressed states, and hence the signal induces rotations of the dressed qubit, regardless to the value of φ (so long as φ is constant during a single experiment).

II. SENSING OF A HIGH FREQUENCY SIGNAL WITH A DOUBLE DRIVE

In order to mitigate the driving fluctuations of the (first) driving field we introduce a second drive, polarized along a perpendicular direction with respect to the polarization of the first driving field. We therefore consider the Hamiltonian

$$H = \frac{\omega_0}{2} \sigma_z + \Omega_1 \left(1 + \delta_{\Omega_1}(t) \right) \cos(\omega_0 t) \sigma_x + \Omega_2 \left(1 + \delta_{\Omega_2}(t) \right) \cos(\omega_0 t) \cos(\Omega_1 t) \sigma_y + g \cos(\omega_s t + \varphi) \sigma_x + \delta_B(t) \sigma_z, \quad (\text{S5})$$

where Ω_2 is the Rabi frequency of the second drive, and we tune the system such that $\omega_s = \omega_0 + \Omega_1 + \frac{\Omega_2}{2}$. In addition, we assume that $\omega_0 \gg \Omega_1 \gg \Omega_2 \gg g$. In the first IP with respect to $H_0 = \frac{\omega_0}{2}\sigma_z$, and after making the RWA and moving to the basis of the dressed states ($x \rightarrow z, z \rightarrow -x, y \rightarrow y$) we have that

$$H_I = \frac{\Omega_1}{2} \left(1 + \delta_{\Omega_1}(t)\right) \sigma_z + \frac{\Omega_2}{2} \left(1 + \delta_{\Omega_2}(t)\right) \cos(\Omega_1 t) \sigma_y + \frac{g}{2} \left(\cos \left(\left(\Omega_1 + \frac{\Omega_2}{2} \right) t + \varphi \right) \sigma_z + \sin \left(\left(\Omega_1 + \frac{\Omega_2}{2} \right) t + \varphi \right) \sigma_y \right) - \delta_B(t) \sigma_x. \quad (S6)$$

We continue by moving to the second IP with respect to $H_{01} = \frac{\Omega_1}{2}\sigma_z$ (in the basis of the dressed states) and taking the RWA to obtain

$$H_{II} = \frac{\delta_{\Omega_1}(t)\Omega_1}{2} \sigma_z + \frac{\Omega_2}{4} \left(1 + \delta_{\Omega_2}(t)\right) \sigma_y + \frac{g}{4} \left(\sigma_+ e^{-i\left(\frac{\Omega_2}{2}t + \varphi\right)} + \sigma_- e^{+i\left(\frac{\Omega_2}{2}t + \varphi\right)} \right), \quad (S7)$$

where we neglected the (fast rotating) terms of the magnetic noise. In the basis of the doubly-dressed states, the σ_y eigenstates ($y \rightarrow z, z \rightarrow -y, x \rightarrow x$), the Hamiltonian is given by

$$H_{III} = \frac{\Omega_2}{4} \left(1 + \delta_{\Omega_2}(t)\right) \sigma_z - \frac{\delta_{\Omega_1}(t)\Omega_1}{2} \sigma_y + \frac{g}{4} \left(\cos \left(\frac{\Omega_2}{2}t + \varphi \right) \sigma_x + \sin \left(\frac{\Omega_2}{2}t + \varphi \right) \sigma_z \right). \quad (S8)$$

Finally, in the third IP with respect to $H_{02} = \frac{\Omega_2}{4}\sigma_z$ (in the basis of the doubly-dressed states), and after making the RWA we have that

$$H_{III} = \frac{g}{8} \left(\sigma_+ e^{-i\varphi} + \sigma_- e^{+i\varphi} \right) + \frac{\delta_{\Omega_2}(t)\Omega_2}{4} \sigma_z, \quad (S9)$$

where we neglected the (fast rotating) terms of the driving fluctuations of Ω_1 , which now contribute only as a second order effect. We therefore conclude that the signal in H_{III} corresponds to the on-resonance coupling between the doubly-dressed states, and hence the signal induces rotations of the doubly-dressed qubit. The doubly-dressed states are vulnerable to fluctuations of Ω_2 , but since $\Omega_2 \ll \Omega_1$ these fluctuations have a smaller effect than fluctuations in Ω_1 ; with $\delta_{\Omega_1}(t) \sim 0.1 - 1\%$ the coherence time is improved by ~ 1 order of magnitude with respect to the coherence time of the dressed states (an improvement of ~ 2 orders of magnitude with respect to T_2^*). In principle, the robustness to driving fluctuations can be further improved by the concatenation of more driving fields [2].

III. IMPROVED SCHEME

In our scheme, a weak signal may further prolong the coherence time of the probe qubit. To see this we continue from equation (S9) and assume for simplicity that $\varphi = 0$. In this case

$$H_{III} = \frac{g}{8} \sigma_x + \frac{\delta_{\Omega_2}(t)\Omega_2}{4} \sigma_z. \quad (S10)$$

However, the difference between the signal, $g \cos(\omega_s t + \varphi) \sigma_x$, and a concatenated third drive, which in our case could be given by $\Omega_3 \cos(\omega_0 t) \cos\left(\frac{\Omega_2}{2}t\right) \sigma_x$, comes from the counter-rotating terms that we usually neglect when making the RWA. Although that the condition $\Omega_1 \gg \Omega_2 \gg g$ holds, examination of the counter-rotating terms of the signal g , the magnetic noise $\delta_B(t)$, and the driving noise of the first drive $\delta_{\Omega_1}(t)$, in the third IP, reveals that there are counter-rotating terms of g and $\delta_B(t)$ and of g and $\delta_{\Omega_1}(t)$ that have identical frequencies. This implies that these terms result in an effective time-independent Hamiltonian [3] that should be included in equation (S10), and hence,

$$H_{III} = \frac{g}{8} \sigma_x + \frac{\delta_{\Omega_2}(t)\Omega_2}{4} \sigma_z + \frac{g\delta_B(t)}{8\Omega_1} \sigma_x + \frac{g\delta_{\Omega_1}(t)}{4\Omega_2} \sigma_x. \quad (S11)$$

The added effective terms imply dephasing rates of $\Gamma_{\delta_B(t)} = \frac{g}{8\Omega_1} S_{BB}(0)$ and $\Gamma_{\delta_{\Omega_1}(t)} = \frac{g}{4\Omega_2} S_{\Omega_1\Omega_1}(0)$, where $S_{BB}(\omega)$ and $S_{\Omega_1\Omega_1}(\omega)$ are the power spectra of the magnetic noise and the noise of the first driving field, respectively. Hence, increasing values of g result in increasing dephasing rates. This was verified in simulations (see Sec. VIII), and to some extent

experimentally as shown in Fig. S5 and in Fig. 4 of main text. Indeed, for small enough values of g this second-order effect is negligible and the signal prolongs the coherence time. For large values of g the coherence time is decreased, even below the coherence time of the doubly-dressed states.

In order to circumvent this problem, we tune the system such that $\omega_s = \omega_0 + \frac{\Omega_2}{2}$ instead of $\omega_s = \omega_0 + \Omega_1 + \frac{\Omega_2}{2}$, which means that we now utilize a different transition in order to couple between the doubly-dressed states (see Fig. S1b). In this case, in the first IP with respect to $H_0 = \frac{\omega_0}{2} \sigma_z$, and after making the RWA and moving to the basis of the dressed states we have that

$$H_I = \frac{\Omega_1}{2} \left(1 + \delta_{\Omega_1}(t)\right) \sigma_z + \frac{\Omega_2}{2} \left(1 + \delta_{\Omega_2}(t)\right) \cos(\Omega_1 t) \sigma_y + \frac{g}{2} \left(\cos\left(\frac{\Omega_2}{2} t + \varphi\right) \sigma_z + \sin\left(\frac{\Omega_2}{2} t + \varphi\right) \sigma_y \right) - \delta_B(t) \sigma_x. \quad (\text{S12})$$

Moving now to the second IP with respect to $H_{01} = \frac{\Omega_1}{2} \sigma_z$ (in the basis of the dressed states) and taking the RWA we obtain

$$H_{II} = \frac{\delta_{\Omega_1}(t) \Omega_1}{2} \sigma_z + \frac{\Omega_2}{4} \left(1 + \delta_{\Omega_2}(t)\right) \sigma_y + \frac{g}{2} \cos\left(\frac{\Omega_2}{2} t + \varphi\right) \sigma_z, \quad (\text{S13})$$

where we neglected the (fast rotating) terms of the magnetic noise. In the basis of the doubly-dressed states, the σ_y eigenstates ($y \rightarrow z, z \rightarrow -y, x \rightarrow x$), the Hamiltonian is given by

$$H_{II} = \frac{\Omega_2}{4} \left(1 + \delta_{\Omega_2}(t)\right) \sigma_z - \frac{\delta_{\Omega_1}(t) \Omega_1}{2} \sigma_y - \frac{g}{2} \cos\left(\frac{\Omega_2}{2} t + \varphi\right) \sigma_y. \quad (\text{S14})$$

Finally, in the third IP with respect to $H_{02} = \frac{\Omega_2}{4} \sigma_z$ (in the basis of the doubly-dressed states), and after making the RWA we have that

$$H_{III} = \frac{g}{4} \left(i \sigma_+ e^{-i\varphi} - i \sigma_- e^{+i\varphi} \right) + \frac{\delta_{\Omega_2}(t) \Omega_2}{4} \sigma_z, \quad (\text{S15})$$

Here, the calculation of the effective Hamiltonian of the counter-rotating terms yields (taking $\varphi = 0$ again)

$$H_{III} = -\frac{g}{4} \sigma_y + \frac{\delta_{\Omega_2}(t) \Omega_2}{4} \sigma_z + \frac{g \delta_B(t)}{8 \Omega_1} \sigma_z. \quad (\text{S16})$$

In this case the second order contribution of the driving noise, $g \delta_{\Omega_1}(t)$, vanishes, and because the second order contribution of the magnetic noise, $g \delta_B(t)$, is perpendicular to the signal, it contributes in higher orders only (assuming $\Omega_1 \gg |\delta_B(t)|$). Therefore, the improved scheme results in even prolonged coherence times and enables the measurement of stronger signals. Moreover, the measured signal in this scheme is stronger by a factor of 2 compared to the original scheme. A comparison between the schemes is shown in Sec. VIII where the results of the simulations suggest that the improved scheme may increase the coherence time by an additional order of magnitude.

IV. DETAILED LEVEL SCHEME AND SETUP

The diamond used for this measurements contains natural abundance of ^{13}C (1.1%) and the selected NV was situated roughly $2\ \mu\text{m}$ below the surface. All the measurements are performed at a static magnetic field of $B_{\text{bias}} = 446\ \text{G}$, where the Nitrogen nuclear spin of the NV is polarized [4], so that the hyperfine transitions appearing from the coupling to the ^{14}N do not intervene with the protocols. The static magnetic field is aligned parallel to the quantization axis of the NV center (connecting the nitrogen and the vacancy) and is defined as the z-axis of the system. Fig. S1a shows an optically detected magnetic resonance (ODMR) measurement from where ω_0 is determined. A more detailed level scheme containing the effect of two

Figure S1. Characterization of the NV system by probing the TLS and a detailed level scheme of the double drive. (a) A pulsed ODMR measurement, where the pulse length of the applied microwave corresponds to $\tau_{\Omega_1}/2 \approx 1.08\ \mu\text{s}$ and the frequency was swept. If the right energy gap ω_0 between the TLS is hit by the microwave, an effective transfer of population from the bright to the dark state can occur which becomes visible in a frequency dependent decrease in fluorescence counts. (b) A detailed level scheme of the double drive procedure with the appearing energy gaps and the driven transitions. Note that in the modulation of the second drive, Ω_2 , a $\pi/2$ was inserted to make Ω_2 not visible in the readout (cf. equation (1) in the main text).

applied drives to the TLS of the NV center is depicted in Fig. S1b.

By switching on the first drive Ω_1 a dressed state configuration is obtained, where the new eigenstates are separated by Ω_1 . In the single drive configuration the energy levels ω_1 and ω_2 are susceptible to external signal. The dressed states are decoupled from external magnetic noise δB , but suffer mainly from the drive noise $\delta\Omega_1$. To decouple the sensor from $\delta\Omega_1$ noise, a second drive Ω_2 of the order of $\delta\Omega_1$ is applied, which drives effectively the appearing transitions ω_1 and ω_2 due to

$$\sin\left(\omega_0 + \frac{\pi}{2}\right) \sin\left(\Omega_1 + \frac{\pi}{2}\right) = \frac{1}{2} \left(\cos(\omega_0 - \Omega_1) + \cos(\omega_0 + \Omega_1) \right) \quad (\text{S17})$$

leading to a doubly-dressed state configuration. Here, four new energy gaps $\omega_s = \omega_0 \pm \omega_1 \pm \omega'_2$ are opened, which can be addressed by an external signal. Note that due to a change in interaction pictures $\omega'_2 = \omega_2/2$. Consequently, the doubly-dressed state suffers mainly from noise contributions of $\delta\Omega_2$.

This concatenation of drives can be continued and will increase the coherence time of the sensor until the noise $\delta\Omega_i$ of the additional applied i-th drive field is on the order of the drive Ω_i itself, or until T_1 time of the bare states of the sensor is reached.

Both drives, Ω_1 and Ω_2 , are sampled and outputted from one channel of an AWG (arbitrary waveform generator, Keysight M8195A) with a time resolution of 65 GS/s. The DAC (digital analog converter) in this device has a resolution of 8bit per channel at maximum 1V (peak-to-peak). As an external signal source, the Rohde&Schwarz SMIQ03B was used.

V. CALCULATING THE SLOPE

The sensor obtains a phase ϕ during a time t which can be written as

$$\phi(t) = \int_0^t \gamma_{\text{NV}} \alpha B d\tau = \gamma_{\text{NV}} \alpha B t \quad (\text{S18})$$

if B remains constant over time. α is a constant factor, which depends on the measurement scheme (and determines the rate of phase accumulation). For a measurement of B under single drive we can set $\alpha_{\text{sd}} = 1/2$ and for the double drive case $\alpha_{\text{dd}} = 1/4$. This can be seen in equation (S9), where the rate at which the signal is recorded is reduced in the double drive case to $(g/2) \cdot \alpha_{\text{dd}} = g/8$.

In the measurement a normalized fluorescence signal S is recorded, where the state $\phi = 0$ is associated with the bright state of the NV (which corresponds to a normalized fluorescence value b). The state $\phi = \pi$ is the dark state of the NV (which corresponds to a normalized fluorescence value d). Therefore the signal accumulation corresponds to

$$S = \frac{(b+d)}{2} + \frac{(b-d)}{2} \cos(2\phi). \quad (\text{S19})$$

since for a TLS we have $S \propto \cos^2(\phi)$.

An unknown magnetic field amplitude B can be extracted by performing a measurement of S . The uncertainty δS of the signal S will eventually determine the error in δB , which are connected by

$$\delta S = \frac{\partial S}{\partial B} \delta B. \quad (\text{S20})$$

The sensor is the most sensitive to a small change in the magnetic field at the point where the signal S has the maximum change, which is the maximal slope

$$\begin{aligned} \max \left| \frac{\partial S}{\partial B} \right| &= \max \left| -\frac{(b-d)}{2} \sin(2\gamma_{\text{NV}} \alpha B \tau) \cdot 2\gamma_{\text{NV}} \alpha \tau \right| \\ &= (b-d) \gamma_{\text{NV}} \alpha \tau = \omega_S C \end{aligned} \quad (\text{S21})$$

Here the full amplitude $C = (b-d)$ of the signal and the rate of change $\omega_S = \gamma_{\text{NV}} \alpha \tau$ in S were introduced. Since S represents a normalized value (normalized with the bright state of the sensor) C directly corresponds to the contrast of the signal. Note that the time (for one measurement run) τ will determine the magnitude of the slope.

To obtain the state of the NV we will count photons. Since the number of photons in one measurement run τ is very small for a NV, we will have to repeat the measurement $N = t/\tau$ times. Each photon record becomes an independent measurement and the uncertainty of the state signal δS is poissonian distributed, $\delta S(t) = 1/\sqrt{N_{ph} \cdot N} = \sigma(t)$. N_{ph} represent the amount of photons counted in one experimental run τ , which can be understood as $N_{ph} = \Gamma_c \tau$, with Γ_c being the count rate.

In the measurement we will accumulate a signal S for a time t and determine the standard deviation $\sigma(t)$ of the accumulated signal over time. Therefore the minimum resolvable magnetic field writes

$$\delta B_{\min}(t, \tau) = \frac{\delta S}{\max \left| \frac{\partial S}{\partial B} \right|} = \frac{1}{\gamma_{\text{NV}}} \frac{\sigma(t)}{\alpha \tau C} = \frac{1}{\gamma_{\text{NV}}} \frac{1}{\alpha C} \frac{1}{\sqrt{N_{ph} \tau t}} \quad (\text{S22})$$

The sensitivity η for a repetitive measurement at τ after time t (i.e. after repeating the measurement $N = t/\tau$ times) is consequently

$$\eta(\tau) = \delta B_{\min}(t, \tau) \sqrt{t} = \frac{1}{\gamma_{\text{NV}}} \frac{1}{\alpha C} \frac{1}{\sqrt{N_{ph} \tau}} \quad (\text{S23})$$

As the signal of the sensor decays with t resulting in a reduced contrast C , the optimal measurement point τ will be situated at the coherence time T_2 , where also the contrast C of the signal should be determined and we will end up in

$$\eta(T_2) = \frac{\hbar}{g\mu_B} \frac{1}{\alpha C} \frac{1}{\sqrt{N_{ph} T_2}} \quad (\text{S24})$$

where $\gamma_{NV} = g\mu_B/\hbar$ was used. A similar derivation can be found in [5]. Note that in general the contrast $C = C(T_2)$ of the signal is depending on the coherence time of the sensor, which mean that it will decay for a time $t > T_2$, which will cause a worst sensitivity η . The appearing constants $1/(\alpha C)$ are measurement related values.

Fig. S2a shows a measurement to determine the maximal slope $|\partial S/\partial B|$ in the double drive case, where an externally applied signal was varied in strength g . With this result it is possible to obtain via $\delta B_{\min} = (2\pi\delta g_{\min})/\gamma_{NV}$ directly $\delta B_{\min}(t, \tau)$ by measuring at the point of the maximal slope $\sigma(t)$ over time t . The result is plotted in Fig. 3 of the main text.

Figure S2. Two different approaches to determine δB_{\min} . (a) Direct slope measurement in double drive at $\tau = 250 \mu\text{s}$ by varying the applied signal strength g . From this measurement the maximal slope $|\partial S/\partial B|_{\max}$ can be obtained. (b) Coherence time measurement in double drive with a signal g . A signal of strength $g/2\pi \approx 41 \text{ kHz}$ was applied during a double drive measurement with $\Omega_1/2\pi = 3.366 \text{ MHz}$ and $\Omega_2/2\pi = 519.5 \text{ kHz}$. In the double drive $g'' = g/4$ is measured. The total measurement took about 1 day (where a signal-to-noise ratio of ≈ 66 was obtained).

Alternatively, it is possible to obtain $\delta B_{\min}(t, \tau)$ by measuring the coherence time under drive with a signal, $\tau = T_2^{\Omega_1, \Omega_2, g}$, and the resulting contrast, C , at the time $T_2^{\Omega_1, \Omega_2, g}$ like in Fig. S2b, to obtain the denominator of equation (S22).

VI. DETERMINE THE OPTIMAL DRIVE PARAMETERS

A. Optimal drive parameter in single drive

Before an external high frequency signal can be measured suitable values for the drive fields Ω_1 and Ω_2 have to be chosen, to obtain the maximal coherence time of the sensor. Since the first drive Ω_1 increases the decoupling of the sensor from the environment but introduces drive noise $\delta\Omega_1$ with a stronger drive Ω_1 , an optimal value has to be selected. The optimal value should prolong the coherence time of the sensor as much as possible for a given drive configuration. Fig. S3 shows Rabi (= single drive) measurements with different drive strength, Ω_1 , displayed against the extracted coherence time of the sensor under drive, $T_2^{\Omega_1}$. It becomes obvious that for a slower drive (< 1 MHz) the coherence time of the sensor, $T_2^{\Omega_1}$, is

Figure S3. Measure the coherence time of the qubit under an increasing drive, Ω_1 , (essentially, just a Rabi measurement by increasing the Rabi frequency.)

not increasing monotonically. One of the reason for this is a less efficient decoupling by Ω_1 allowing various interactions between individual nearby ^{13}C nuclear spins and the overall ^{13}C bath contribution. For a static magnetic field of $B = 446$ G and $\gamma_{^{13}\text{C}}/2\pi = 1.0705$ kHz/G we expect the Larmor frequencies of the ^{13}C bath to be around $\nu_{\text{larmor}^{13}\text{C}} = \gamma_{^{13}\text{C}}B \approx 477$ kHz, which fits well with the large dip at around 500 kHz. To use the dressed states as a high frequency sensor under a single drive, we select $\Omega_1/2\pi > 2.5$ MHz based on Fig. S3 to make sure that a sufficient decoupling from the environment can be guaranteed. Compared to the pure dephasing time of this NV with $T_2^* = 1.1$ μ s, the single drive with $\Omega_1/2\pi = 2.5$ MHz improves the coherence time already by about a factor of 60.

B. Optimal drive parameter in double drive

In principle, we could select for the double drive experiments a very strong first drive Ω_1 , but then a stronger second drive Ω_2 is required to correct the noise on the order of $\delta\Omega_1$. Consequently, a balanced first drive Ω_1 (which maximizes Fig. S3) will eventually introduce less drive noise Ω_2 , and prolong the coherence time of the sensor. For a moderate first drive $\Omega_1/2\pi = 3.363\text{MHz}$, the second drive Ω_2 was scanned in Fig. S4 in order to find the optimal choice for Ω_2 . This scan

Figure S4. Set the first drive to be $\Omega_1/2\pi = 3.363\text{MHz}$. Vary the second drive, Ω_2 , and record the coherence time, $T_2^{\Omega_1, \Omega_2}$, of the qubit, where essentially a double drive measurement without an external signal was performed. The optimal second drive, Ω_2 , maximizes the coherence time $T_2^{\Omega_1, \Omega_2}$ of the sensor.

of Ω_2 has to be done, if the exact magnitude or characteristics of $\delta\Omega_1$ is not known.

Note that the measured coherence time in Fig. S2b is longer compared to the the maximal coherence time obtained in Fig. S4. In Sec. VII it is shown that an applied signal with strength, g , acting partially as a third drive, can further increase the coherence time of the sensor for certain values of g .

VII. DETERMINE THE COHERENCE TIME OF THE SINGLE DRIVE UNDER AN INCREASING SIGNAL

To show the impact of an external signal strength g on the coherence time of the sensor under drive, we perform a single drive measurement (corresponding to one point in Fig. S3) by now varying the external signal strength g . We set $\Omega_1/2\pi = 3.002$ MHz and increase gradually g . For each measurement point a decaying trace was recorded, which was (under)sampled at multiples of $\tau_{\Omega_1} = 2\pi/\Omega_1$, i.e. we measure at times $t = N\tau_{\Omega_1}$ ($N \in \mathbb{N}$), which removes the effect of Ω_1 on the recorded traces. Consequently, g' was obtained by fitting the data to an exponential decaying sine function. For small values of g , recording the oscillations of g' , and subsequently the coherence time proved to be a challenging measurement, mainly because the period of g' was longer than the coherence time $T_2^{\Omega_1, g}$, so it was difficult to distinguish the two. This caused the measured coherence time to be shorter than expected, given we would expect the coherence time at small g to converge with the measurements without signal (Fig. S3), thus having larger error bars to compensate the ambiguity. Interestingly, the signal acts to a certain extent as a double drive and prolongs the coherence time which is pointed out in Sec. III. Comparing Fig. S4 and Fig. S5

Figure S5. Set the first drive to be $\Omega_1/2\pi = 3.002$ MHz and vary the applied external signal (which is not phase locked to the sensor). Display of the qubit's coherence time at a fixed drive, Ω_1 , and with an increasing signal strength g .

two main differences arise. First, the maximal coherence times in Fig. S5 is reached at a different ratio ($g/\Omega_1 \approx 0.0083$) and second, the maximal coherence time at this point is about the half it was by using Ω_2 . This can be explained by the following. The signal g is not phase locked to the sensor, therefore we would expect that on average $\sum G(t) = \sum g \cos^2(\omega_s t + \langle \varphi \rangle) \approx \frac{1}{2}g$ is contributing to the sensor. A phase locked signal, however, will completely contribute to the decoupling with g , since φ does not vary between each measurement run, which justifies the discrepancy on the y axis in the graphs.

Furthermore, it seems that Ω_2 introduces more noise to the system than g does. As a consequence, $\delta\Omega_2$ is larger and a stronger second drive is needed to cope for the resulting overall noise level.

On the basis of a very rough estimation, we can give some upper limits for the amplitude noise of the microwave fields, which are mentioned in Sec. IV. For the maximal output range of 1 Vpp we can assume $1 \text{ Vpp}/2^8 \approx 0.004 \text{ Vpp}$ to be the absolute noise level for this device (Keysight M8195A). Ω_1 was created by 0.75 Vpp and based on Fig. S4 the optimal second drive has to be $\Omega_2 = 0.15\Omega_1 = 0.1125 \text{ Vpp}$. Eventually, Ω_2 suffers from a larger relative noise level ($0.004 \text{ Vpp}/0.1125 \text{ Vpp} \approx 3.5\%$) then the first drive Ω_1 ($0.004 \text{ Vpp}/0.75 \text{ Vpp} \approx 0.4\%$). This statement can be further confirmed by comparing the relative noise $\delta\Omega_2/\Omega_2$ with $\delta g/g$. The external signal was produced by a microwave generator (R&S SMIQ03B), which runs continuously during the measurements and has a more than 100 times cleaner signal (-70 dBc, relative amplitude noise ratio 0.0003), compared to $\delta\Omega_2/\Omega_2$. This indicates also why the coherence time in Fig. 2 of the main text could be prolonged with g by such a great amount, since the relative (and absolute) noise of the signal g seems to be much smaller than $\delta\Omega_2$. Eventually, a coherence

time of 1.5 ms should be almost the limit to which we could extend the coherence time, since T_1 of the sensor was about 3 ms.

VIII. SIMULATIONS

In this section we present the results of simulations aimed at reproducing the experimental results and verifying our theoretical model.

Figure S6. Simulations of coherence times. (a) Oscillations between $|\uparrow_x\rangle$ and $|\downarrow_x\rangle$ averaged over 5000 trails. $(1 + e^{-\frac{g^2 t^2}{2}})/2$ is plotted in green ($g^2 = 2/(T_2^*)^2$). (b) Probability of being in the initial $|\uparrow_x\rangle$ state averaged over 2400 trails. Hahn echo pulse at $t = 515/2$ μs and refocusing at $T_2 = 515$ μs. Dashed horizontal line at $P = (1 + 1/e)/2$. (c) Coherence time under a single drive with a Rabi frequency of $\Omega_1 = 2\pi \cdot 3.4$ MHz. Oscillations between $|\uparrow_y\rangle$ and $|\downarrow_y\rangle$ averaged over 800 trails. Dashed horizontal line at $P = (1 + 1/e)/2$. The simulation indicates a coherence time of $T_2^{\Omega_1} \approx 33$ μs. (d) Coherence time under a double drive with Rabi frequencies of $\Omega_1 = 2\pi \cdot 3.4$ MHz and $\Omega_2 = 0.15 \cdot \Omega_1$. Oscillations between $|\uparrow_x\rangle$ and $|\downarrow_x\rangle$ averaged over 800 trails. Dashed horizontal line at $P = (1 + 1/e)/2$. The simulation indicates a coherence time of $T_2^{\Omega_1, \Omega_2} \approx 380$ μs.

A. Magnetic noise

The NV center spin used in our experiment had a pure dephasing time of $T_2^* = 1.1 \mu\text{s}$, and under a Hahn Echo pulse, a coherence time of $T_2 = 515 \mu\text{s}$. In our model, the magnetic noise has two components. One component, B_r , is a random field that is static within each experiment, but has a different value in different experiments, and a second component, $B(t)$, which is an Ornstein-Uhlenbeck process [6, 7] with a zero expectation value, $\langle B(t) \rangle = 0$, and a correlation function $\langle B(t)B(t') \rangle = \frac{c\tau}{2} e^{-\gamma|t-t'|}$. B_r is normally distributed with a variance of $\sigma^2 \approx \frac{0.96 \times 2}{T_2^*}$, where $T_2^* = 1.1 \mu\text{s}$. An exact simulation algorithm [8] was employed to realize the Ornstein-Uhlenbeck process, which according to

$$B(t + \Delta t) = B(t)e^{-\frac{\Delta t}{\tau}} + n\sqrt{\frac{c\tau}{2} \left(1 - e^{-\frac{2\Delta t}{\tau}}\right)}, \quad (\text{S25})$$

where n is a unit Gaussian random number. The correlation time of the noise was set to $\tau = 1/\gamma = 10 \mu\text{s}$, where the diffusion constant is given by $c \approx \frac{2}{T_2^*\tau}$ and corresponds to $S_{BB}(0) \approx \frac{1}{515}$. For pure dephasing, we simulated the Hamiltonian

$$H = \frac{\omega_0}{2} \sigma_z + (B_r + B(t)) \sigma_z, \quad (\text{S26})$$

where we used $\omega_0 = 100 \text{ MHz}$ and the qubit was initialized to $|\uparrow_x\rangle$. The result of the simulation is shown in Fig. S6a and corresponds to a pure dephasing time of $T_2^* = 1.1 \mu\text{s}$.

For the Hahn echo pulse we run the same simulation but flipped the sign of $(B_r + B(t)) \sigma_z$ at $t = \frac{515}{2} \mu\text{s}$. The result is shown in Fig. S6b and corresponds to a Hahn echo time of $T_2 = 515 \mu\text{s}$.

B. Single drive

We simulated the single-drive scenario under the RWA, so the Hamiltonian is given by

$$H = \frac{\Omega_1}{2} \left(1 + \delta_{\Omega_1}(t)\right) \sigma_x + (B_r + B(t)) \sigma_z. \quad (\text{S27})$$

Driving fluctuations were also modelled by an Ornstein-Uhlenbeck process with a zero expectation value. We chose a correlation time of $\tau_{\Omega_1} = 500 \mu\text{s}$, and a relative amplitude error of $\delta_{\Omega_1} = 0.15\%$ so the diffusion constant is given by $c_{\Omega_1} = 2\delta_{\Omega_1}/\tau_{\Omega_1}$. We set $\Omega_1 = (2\pi) \cdot 3.4 \text{ MHz}$, and the qubit was initialized to $|\uparrow_y\rangle$. The result of the simulation is shown in Fig. S6c and indicates a coherence time of $T_2^{\Omega_1} \approx 33 \mu\text{s}$, which is in agreement with the experimental results.

C. Double drive

We simulated the double drive scenario in the first IP, making the RWA with respect to ω_0 only. The Hamiltonian is given by

$$H = \frac{\Omega_1}{2} \left(1 + \delta_{\Omega_1}(t)\right) \sigma_x + \frac{\Omega_2}{2} \left(1 + \delta_{\Omega_2}(t)\right) \cos(\Omega_1 t) \sigma_y + (B_r + B(t)) \sigma_z. \quad (\text{S28})$$

Due to some technical issues, in our experimental system the noise in Ω_2 was stronger than the noise in Ω_1 . We therefore set $\delta_{\Omega_1} = 0.15\%$ and $\delta_{\Omega_2} = 0.21\%$, where $\Omega_1 = 2\pi \cdot 3.4 \text{ MHz}$ (like in the single-drive case) and $\Omega_2 = 0.15\Omega_1$. The spin was initialized to $|\uparrow_x\rangle$. The result of the simulation is shown in Fig. S6d and indicates a coherence time of $T_2^{\Omega_1, \Omega_2} \approx 380 \mu\text{s}$, which is in agreement with the experimental results.

D. Double drive with a signal

In this section we used the experimental parameters used in the experiment of Fig. S2b, where a signal of $g/2\pi = 41.084 \text{ kHz}$ is sensed by the doubly-dressed qubit. The simulation result is shown in Fig. S7a and indicates a coherence time of $T_{2,\text{sim}}^{\Omega_1, \Omega_2} \approx$

450 μs , which is shorter than the experimental value of $T_{2,\text{exp}}^{\Omega_1, \Omega_2} \simeq 600 \mu\text{s}$. This could be because our theoretical model of the magnetic noise does not provide a complete characterization of the actual noise, and in fact describes a more severe situation. In principle, the actual spectrum of the noise can be measured [9], which would allow for a theoretical optimization of the parameters used in the sensing protocol. We then varied the value of g in order to verify its effect on the coherence time. For Fig. S7b (Fig. S7c) we decreased (increased) the value of g by a factor of 2, and used $g/2\pi = 0.5 \cdot 41.084 \text{ kHz}$ ($g/2\pi = 2 \cdot 41.084 \text{ kHz}$). For the smaller (larger) value of g the coherence time was increased (decreased), as expected. We then used the original g value of $g/2\pi = 41.084 \text{ kHz}$, and simulated the improved scheme with $\omega_i = \omega_0 + \frac{\Omega_2}{2}$. The result of this simulation is shown in Fig. S7d. The simulation confirms the improved performance of the scheme, and indicates an improvement of the coherence time by ~ 1 order of magnitude.

Figure S7. High-frequency signal sensing by a double drive. Dashed horizontal line at $P = (1 \pm 1/e)/2$. (a) Simulation of high-frequency signal sensing by a double drive with the experimental parameters used in the experiment of Fig. S2b. (b) A weaker signal increases the coherence time. All parameters are identical to those of Fig. S7a except g , which is decreased by a factor of 2. (c) A stronger signal decreases the coherence time. All parameters are identical to those of Fig. S7a except g , which is increased by a factor of 2. (d) Improved scheme. All parameters are identical to those of Fig. S7a.

IX. PULSED ANALOG OF THE SCHEME

The sensing of high frequency fields with a TLS could also be achieved with a pulsed dynamical decoupling analog of our scheme. This could be seen by considering an AC signal with a frequency of $\omega_s = \omega_0 + \Omega$ (similar to the single-drive case). The Hamiltonian of the system under the AC signal is given by

$$H = \frac{\omega_0}{2} \sigma_z + g \sigma_x \cos(\omega_s t), \quad (\text{S29})$$

and in the interaction picture with respect to $H_0 = \frac{\omega_0}{2} \sigma_z$ we have that

$$\begin{aligned} H_I &= \frac{g}{2} \left(\sigma_+ e^{-i\Omega t} + \sigma_- e^{+i\Omega t} \right) \\ &= \frac{g}{2} (\cos(\Omega t) \sigma_x + \sin(\Omega t) \sigma_y). \end{aligned} \quad (\text{S30})$$

Hence, the signal can be measured by initializing the spin to $|\uparrow_x\rangle$ and applying a CPMG pulse sequence, where the pulses correspond to π rotations around the \hat{y} axis (so the σ_y part of the signal is measured). Similar to the common pulsed dynamical decoupling sensing methods, the rate of the pulses should match Ω in order for the signal g to be observed.

-
- [1] N. Aharon, I. Cohen, F. Jelezko, and A. Retzker, *New Journal of Physics* **18**, 123012 (2016).
 - [2] J.-M. Cai, B. Naydenov, R. Pfeiffer, L. P. McGuinness, K. D. Jahnke, F. Jelezko, M. B. Plenio, and A. Retzker, *New Journal of Physics* **14**, 113023 (2012).
 - [3] D. F. James and J. Jerke, *Canadian Journal of Physics* **85**, 625 (2007).
 - [4] V. Jacques, P. Neumann, J. Beck, M. Markham, D. Twitchen, J. Meijer, F. Kaiser, G. Balasubramanian, F. Jelezko, and J. Wrachtrup, *Physical Review Letters* **102**, 057403 (2009).
 - [5] L. M. Pham, *Magnetic Field Sensing with Nitrogen-Vacancy Color Centers in Diamond*, *Doctor of Philosophy*, Harvard University, Cambridge, Massachusetts (2013).
 - [6] M. C. Wang and G. E. Uhlenbeck, *Reviews of Modern Physics* **17**, 323 (1945).
 - [7] R. Hanson, V. V. Dobrovitski, A. E. Feiguin, O. Gywat, and D. D. Awschalom, *Science* **320**, 352 (2008).
 - [8] D. T. Gillespie, *Physical Review E* **54**, 2084 (1996).
 - [9] Y. Romach, C. M{a}ller, T. Uden, L. Rogers, T. Isoda, K. Itoh, M. Markham, A. Stacey, J. Meijer, S. Pezzagna, B. Naydenov, L. McGuinness, N. Bar-Gill, and F. Jelezko, *Physical Review Letters* **114**, 017601 (2015).

REVIEWERS' COMMENTS:**Reviewer #2 (Remarks to the Author):**

The authors have done a good job in replying to my original criticism. I like the fact that the main differences between this work and Ref. 36 are now clearly outlined in the main text; this really helps the reader. Going beyond the t_2^* limit in sensing is an extremely valuable challenge. In addition, with all the other corrections, I believe this manuscript is now a valuable piece of work and it deserves publication in Nature Communications.

Reviewer #3 (Remarks to the Author):

The readability of the revised manuscript has been significantly improved from the original submission. Also, the authors have clarified the importance of the present work (and the difference from previous works) in the Introduction part. These are all welcome changes. I think all the issues raised in my previous review have been addressed satisfactorily in the revised manuscript, and I now recommend the present work for publication in Nature Communications.